**Data Availability Statement:** All relevant data are within the manuscript and its Supporting Information files.

# The sexual behaviours of adolescents aged between 14 and 17 years involved with the juvenile justice system in Australia: A community-based survey

**Lorraine Yap**[1], **Jocelyn Jones**[2], **Basil Donovan**[1], **Sally Nathan**[3], **Elizabeth Sullivan**[4], **Sophie Davison**[2,5], **Ed Heffernan**[6,7,8], **Alun Richards**[9], **Carla Meurk**[6,7], **Megan Steele**[7], **Christopher Fisher**[10], **Bianca Ton**[3], **Tony Butler**[3]*

1 The Kirby Institute, The University of New South Wales, Sydney, Australia, 2 National Drug Research Institute, Perth, Western Australia, Australia, 3 School of Population Health, The University of New South Wales, Sydney, Australia, 4 The University of Newcastle, Callaghan, New South Wales, Australia, 5 State Forensic Mental Health Service, North Metropolitan Health Service, & Office of the Chief Psychiatrist, Nedlands, Western Australia, Australia, 6 Queensland Forensic Mental Health Service, Brisbane, Australia, 7 Queensland Centre for Mental Health Research, Brisbane, Australia, 8 The University of Queensland, Brisbane, Australia, 9 Immunisation Program and BBV/STI Unit, Communicable Diseases Branch Queensland Health, Brisbane, Queensland, Australia, 10 Australian Research Centre in Sex, Health and Society, La Trobe University, Melbourne, Australia

* tbutler@unsw.edu.au

## Abstract

### Objectives

To overcome key knowledge gaps in relation to justice involved and vulnerable young people and their sexual health and to compare this group with their peers from other youth health surveys in Australia to determine the extent of the issues.

### Methods

Young people, aged between 14 and 17 years, who had ever been or were currently involved with the criminal justice system were purposively sampled. The survey was anonymous and delivered using Computer Assisted Telephone Interview (CATI).

### Results

A total of 465 justice involved MeH-JOSH young people, aged between 14 and 17 years, participated in the study: 44% Aboriginal and/or Torres Strait Islander (Indigenous) and 37% not attending school. Of the total valid responses, 76% (n = 348) reported having ever had sex, with sexual initiation at a median age of 14 years. We compared these data with their peers in other Australian surveys and found that young people in our study had a higher engagement in sex and start having sex at a younger age, reporting more sexual partners at all ages.

**Funding:** The authors disclosed that they received the following support for their research and/or authorship of this article: MeH-JOSH study was funded by the Australian National Health and Medical Research Council Project Grant No. 1043693 (awarded to TB, LY, BD, ES, SN). Tony Butler is supported by an NHMRC Fellowship and Basil Donovan is supported by a NHMRC Practitioner Fellowship: https://www.nhmrc.gov.au. Any material published or made publicly available by the researchers cannot be considered as either endorsed or an expression of the policies or view by the Western Australia Department of Justice and Western Australia Department of Health. Any errors of omission or commission are the responsibility of the researchers. The funders had no role in study design, data collection and analysis, decision to publish, or preparation of the manuscript.

**Competing interests:** The authors have declared that no competing interests exist.

## Conclusions

The sexual behaviours of young people involved in the justice system in this study suggest they may be at a greater risk for sexually transmissible infections than their age-matched peers in the general population. Policymakers should elevate them to a priority population for targeting sexual health services and health promotion.

## Introduction

Young people involved with the justice system aged between 14–17 years are under-researched in relation to their sexual health in Australia and internationally. They are also neglected by sexual health programmes even though they are potentially one of the most vulnerable groups in society. The lack of health information on young people under youth justice supervision was recently acknowledged by the Council of Australian Governments, the Australian Human Rights Commission, and the Royal Australasian College of Physicians [1]. Limited social, educational and economic opportunities place them at a further disadvantage in reducing the risk of harm to their health [2, 3]. Itinerancy and disengagement from traditional schooling compounds access to this group from a research perspective [4].

Studies have suggested that drug addiction and mental health disorders that put youth at risk for offending may also drive their tendency to engage in riskier sexual behaviours [5–8]. One study reported increased sexual risk behaviours among juvenile detainees compared to the general population [9]. In Australia, a survey was conducted among juvenile justice youth detention in New South Wales. Of 19 young females surveyed, 31.6% had ever been pregnant, first pregnancy was at 14.2 years. No questions were asked on abortions, miscarriages, stillbirths or drugs/smoking/alcohol while pregnant [3]. Formal comparisons of young offenders with community populations and their sexual health and behaviours are extremely limited [10].

In Australia, 83% or 4,568 young people were under juvenile justice community supervision and 18% or 974 young people were in detention on an average day [11]. Community based supervision in Australia includes unsentenced orders (supervised or conditional bail) and sentenced orders (probation and similar order, suspended detention, parole or supervised release) [12].

Young people who do not fall under the formal supervision of the youth justice system but nonetheless who have had contact with the justice system have been neglected from a research perspective due to itinerancy and disengagement from education and other mainstream services (including government sponsored youth justice services), precluding their involvement in mainstream health surveys. These young people may include those who have had police warnings, cautions, fines, good behaviour bonds or who have been diverted to other types of detention such as, a police watch-house or adult prison, or have been diverted to other programs (e.g. flexi-learning centres and colleges, drug and alcohol treatment centres). We know even less about this population.

The Mental Health, Sexual Health and Reproductive Health of Young People in Contact with the Criminal Justice System (MeH-JOSH) study aimed to describe the sexual and mental health and risk behaviours of young people (14–17 year olds) in the community who had ever been in contact with the criminal justice system (e.g. police, youth justice, court system, tribunals). The main aim of the survey was to overcome key knowledge gaps in relation to this population and compare this group with their peers from other health surveys of youth in Australia.

## Methods

### Sampling and sample size

A purposive sampling design based on strict selection criteria was used to recruit young people in the community; participants must be aged between 14 and 17 years and had to have had contact with the criminal justice system in the past or present. This approach was necessary as permission was not granted to conduct the survey with young people in detention and those serving community orders.

A sample of young people aged 14–17 years old was selected to reflect the age of the youth justice population in Australia. Those over 17 (that is, 18 or older) are considered adults. Ethical issues prevented interviewing those below 14 years, such as obtaining parental/guardian consent, and also having to ensure that very young respondents could truly understand what they would be consenting to.

Quota sample sizes were calculated based on known demographic characteristics of the Australian juvenile offender population (age and gender) [11]. Aboriginal and/or Torres Strait Islander (Indigenous) young people in this study sample were over represented (44%) compared to 5% in the general Australian population and this reflects the greater involvement of this population in the justice system [13]. In Australia, the majority of young people under supervision on an average day in 2017–18 were male (81%) [11]. Females were deliberately oversampled in this study to enable more advanced statistical analysis.

### Recruitment

Recruitment took place between June 2016 and August 2018 using four different recruitment strategies and to minimise selection bias. Firstly, young people who met the selection criteria were recruited through referrals by programme coordinators from community-based organisations and youth drop-in centres: 70% of the total participants were recruited using this approach. The second strategy involved recruiting young people waiting inside or outside the magistrate courts on days the Children's Court was in session. A third strategy was to recruit justice involved young people from youth mental health service centres in Western Australia. Programme coordinators referred young people who satisfied the selection criteria and agreed to participate in the study. A fourth strategy was to recruit justice involved young people attending flexi-learning schools or colleges. These institutions aim to re-engage young people who had been out of the formal school system.

### Consent

Due to the nature of this population and from our experiences of conducting surveys among young offenders in Australia [4], the study expected that a high proportion of young people would not have a good relationship with their parent(s) or with the adults responsible for them. Thus human research ethics approval was sought and given for young people recruited in the survey to be treated as mature minors [14]. Nevertheless, all young people approached outside the courts were still asked by recruiters if we could contact their parents or guardians for permission to allow them to participate in the survey. Almost all respondents refused us contact or to give any contact details of their parents or guardians preferring to give consent themselves. Recruiters were required to administer a Gillick Competency checklist to ensure that all respondents met the criteria of a mature minor.

### Survey

Having obtained written consent from the young person, the survey was delivered using a Computer Assisted Telephone Interview (CATI). The average time to complete the survey was

approximately 40 minutes. No identifying information was recorded by the interviewer to ensure anonymity.

Data were collected on the following: socio-demographics; history of justice system involvement; sexual identity, sexual attraction and sexual history; sexual health behaviours and knowledge; human papillomavirus (HPV) vaccination; and sexually transmissible infection (STIs) history. Survey questions were designed to be consistent with other Australian surveys of young people to enable comparisons.

Original datasets of the MeH-JOSH survey were age matched with the Young Minds Matter: 2nd Survey of the Mental Health of Australian Children Survey (YMM), a probability sample of young people aged between 4 and 17 years old from 5,500 randomly sampled families in Australia [15]. Separate age matched data analysis was provided by C. Fisher for the 6th National Survey of Secondary Students and Sexual Health (SSASH), a convenience sample of adolescents in the community that had agreed to take part in an online survey in Australia [16]. MeH-JOSH sexual health and behavioural questions for this paper were selected based on the equivalence or sameness of questions from the YMM and SSASH survey (for example, MEH-JOSH–"Can you tell me if you have ever had sex?" YMM and SSASH–"Have you ever had sexual intercourse?").

After completion of the survey, participants were given either $50 cash or a gift card to reimburse them for their time and travel. No incentives were given to participants recruited inside the Children's Court in Western Australia as it was deemed inappropriate. Consequently no participants from this location agreed to take part in the survey.

**Post-survey interview.**   All participants were given post-survey exit interviews and, if necessary, provided with referrals to relevant health agencies to ensure they were not adversely impacted by the survey questions. We found that HPV vaccinations were the most sought after referral request.

## Participants

A total of 465 justice involved young people (63% male and 37% female), aged between 14 and 17 years, consented to participate in the MeH-JOSH survey. There was a higher proportion of males than females in the 16–17 age group (56% versus 44%, $\chi^2(1) = 5.683$, $p = .017$). Of the total surveyed, 85% were born in Australia, 44% identified as Aboriginal and/or Torres Strait Islanders (Indigenous) and 37% were not attending secondary school or a flexi-school or college (Table 1). A higher proportion of males were on a current sentence or order compared to females (40% versus 30%, $\chi^2(1) = 4.567$, $p = .033$). Site of recruitment was not a confounding factor for age and gender although it was significant for Aboriginality (Indigenous status) ($\chi^2(4) = 12.026$, $p = .017$).

MEH-JOSH survey data was compared to two community surveys of: (i) 2004 age-matched young people from the YMM survey (51% male and 49% female), 89% of whom were attending secondary school or a flexi-school or college, and (ii) 7170 age-matched young people (43% male, 55% female and 2% trans or gender diverse) from the SSASH survey, 5% of whom identified as Aboriginal and/or Torres Strait Islanders (Indigenous), 91% born in Australia and 94% attending secondary school or a flexi-school or college (Table 1).

## Data analysis

### Sexual health and behaviours

Descriptive statistics were generated using Statistics SPSS 25. The self-reported sexual health and behaviours of participants in the MEH-JOSH survey (including sexual identity and attraction, relationship status, ever having sex (any, vaginal, anal, oral), sexual initiation, lifetime

**Table 1. Sociodemographic profiles of community-based justice involved young people (MeH-JOSH) and young people from selected community-based population surveys in Australia, aged between 14 and 17 years.**

| DESCRIPTION | | Justice-involved population | | | | General population | |
|---|---|---|---|---|---|---|---|
| | | Community based | | | | | |
| | | MeH-JOSH | | | | 2014 YMM[1] | 2018 SSASH[2] |
| | | Male | Female | Total | *p* value | Total | Total |
| | | n (%) | n (%) | n (%) | | n (%) | n (%) |
| | | | | 465 | | 2004 | 7170 |
| **Gender** | Male | - | - | 293 (63.0) | - | 1029 (51.3) | 3067 (42.8) |
| | Female | - | - | 172 (37.0) | | 975 (48.7) | 3930 (54.8) |
| | Trans and Gender Diverse | - | - | 0 (0.0) | | 0 (0.0) | 173 (2.4) |
| | **Total** | - | - | **465 (100.0)** | | **2004 (100.0)** | **7170 (100.0)** |
| **Age Group** | 14–15 years | 130 (44.4) | 96 (55.8) | 226 (48.6) | **0.017** | 652 (32.5) | 2046 (28.5) |
| | 16–17 years | 163 (55.6) | 76 (44.2) | 239 (51.4) | | 1352 (67.5) | 5124 (71.5) |
| | **Total** | **293 (100.0)** | **172 (100.0)** | **465 (100.0)** | | **2004 (100.0)** | **7170 (100.0)** |
| **Aboriginal and/or Torres Strait Islander (Indigenous)** | Yes | 124 (42.5) | 78 (45.6) | 202 (43.6) | **0.510** | - | 298 (4.6) |
| | No | 168 (57.5) | 93 (54.4) | 261 (56.4) | | - | 6697 (95.7) |
| | Prefer not to say/Missing* | 1 (-) | 1 (-) | 2 (-) | | - | 175 (-) |
| | **Total** | **292 (100.0)** | **171 (100.0)** | **463 (100.0)** | | - | **7170 (100.0)** |
| **Place of Birth** | Australia | 243 (82.9) | 154 (89.5) | 397 (85.4) | **0.147** | - | 6481 (90.8) |
| | New Zealand | 35 (11.9) | 12 (7.0) | 47 (10.1) | | - | 89 (1.2) |
| | Other | 15 (5.1) | 6 (3.5) | 21 (4.5) | | - | 565 (8.0) |
| | Prefer not to say/Missing* | 0 (-) | 0 (-) | 0 (-) | | - | 35 (-) |
| | **Total** | **293 (100.0)** | **172 (100.0)** | **465 (100.0)** | | - | **7170 (100.0)** |
| **Education** | Goes to school | 184 (62.8) | 110 (64.0) | 294# (63.2) | **0.803** | 1780 (88.8) | 6679 (94.2) |
| | Not at school | 109 (37.2) | 62 (36.0) | 171 (36.8) | | 224 (11.2) | 412 (5.8) |
| | Prefer not to say/Missing* | 0 (-) | 0 (-) | 0 (-) | | 0 (-) | 79 (-) |
| | **Total** | **293 (100.0)** | **172 (100.0)** | **465 (100.0)** | | **2004 (100.0)** | **7170 (100.0)** |
| **Most serious sentence/order** | Community Low–Fines, Bond, Caution, Warning | 130 (44.4) | 85 (49.4) | 215 (46.2) | **0.341** | - | - |
| | Community High–Probation and Community Order | 46 (15.7) | 25 (14.5) | 71 (15.3) | | - | - |
| | Detention–Juvenile Detention, Police Watchhouse, Prison | 104 (35.5) | 50 (29.1) | 154 (33.1) | | - | - |
| | Other Contact–Diversion, Drug court | 13 (4.4) | 12 (7.0) | 25 (5.4) | | - | - |
| | **Total** | **293 (100.0)** | **172 (100.0)** | **465 (100.0)** | | - | - |

(*Continued*)

**Table 1.** (Continued)

| DESCRIPTION | | Justice-involved population | | | | General population | |
|---|---|---|---|---|---|---|---|
| | | Community based | | | | | |
| | | MeH-JOSH | | | | 2014 YMM[1] | 2018 SSASH[2] |
| | | Male | Female | Total | p value | Total | Total |
| | | n (%) | n (%) | n (%) | | n (%) | n (%) |
| | | | | 465 | | 2004 | 7170 |
| **Offence categories** (May have more than one answer) | Theft and related offences | 109 (37.2) | 77 (44.8) | 186 (40.0) | **0.298** | - | - |
| | Unlawful entry with intent/burglary, break and enter | 54 (18.4) | 13 (7.6) | 67 (14.4) | **0.344** | - | - |
| | Robbery, extortion and related offences | 46 (15.7) | 9 (5.2) | 55 (11.8) | **0.406** | - | - |
| | Acts intended to cause injury | 63 (21.5) | 53 (30.8) | 116 (24.9) | **0.254** | - | - |
| | Public order offences | 41 (14.0) | 22 (12.8) | 63 (13.5) | **0.895** | - | - |
| | Property damage and environmental pollution | 50 (17.1) | 5 (2.9) | 55 (11.8) | **0.407** | - | - |
| | Illicit drug offences | 32 (10.9) | 12 (7.0) | 44 (9.5) | **0.699** | - | - |
| | Offences against government procedures, security and operations | 19 (6.5) | 10 (5.8) | 29 (6.2) | **0.941** | - | - |
| | Traffic and vehicle regulatory offences | 16 (5.5) | 7 (4.1) | 23 (4.9) | **0.888** | - | - |
| | Abduction, harassment and other offences against the person | 8 (2.7) | 5 (2.9) | 13 (2.8) | **0.980** | - | - |
| | Fraud, deception and related offences | 6 (2.0) | 1 (0.6) | 7 (1.5) | **0.922** | - | - |
| | Sexual assault and related offences | 2 (0.7) | 2 (1.2) | 4 (0.9) | **0.959** | - | - |
| | Other | 9 (3.1) | 3 (1.8) | 12 (2.6) | **0.906** | - | - |
| **On a current order** | Yes | 116 (40.1) | 51 (30.2) | 167 (35.5) | **0.033** | - | - |
| | No | 173 (59.9) | 118 (69.8) | 291 (63.5) | | - | - |
| | Prefer not to say/Missing* | 4 (-) | 3 (-) | 7 (-) | | - | - |
| | **Total** | **289 (100.0)** | **169 (100.0)** | **458 (100.0)** | | - | - |

* Prefer not to say/Missing not included in the total n (%) and p-value calculations

[1] 2014 YMM– 2014 Young Minds Matter: 2nd Survey of the Mental Health of Australian Children Survey

[2] 2018 SSASH– 6th National Survey of Secondary Students and Sexual Health

#Includes secondary high schools and flexi- (flexible) schools and colleges

sexual encounters, last sexual encounter, STI testing and diagnosis, HPV vaccination, and willingness to have HPV vaccine) were compared with their age-matched peers in the YMM [15] and SSASH [16] surveys where applicable. Chi-squared tests were used to analyse gender differences in these variables in the MEH-JOSH survey. A test of two proportions was used to analyse gender differences in types of offences committed.

## Ethics

Approval for the study was obtained from the University of New South Wales Sydney Human Research Ethics Committee, (HC13308), the Western Australia Aboriginal Health Ethics Committee (WAAHEC 625), and Curtin University (HRE0133). Permission was also granted in Western Australia by the North Metropolitan Health Service Mental Health Research Ethics Committee (22_2016), the North Metropolitan and East Metropolitan Health Services' Research Governance, and the Department of the Justice Research Application Advisory Committee (ref 2016/02161) to recruit from their respective premises.

## Results

### Sexual health and behaviours

Of the MeH-JOSH participants, 91% identified as heterosexual, 6% bisexual, and 1% homosexual/lesbian. Young females in the MeH-JOSH study were more likely to identify as bisexual than young males, (15% versus 1%, $\chi^2(3) = 34.295$, $p < .001$), and this is also reflected in their sexual attraction (14% versus 3%, $\chi^2(3) = 20.031$, $p < .001$). This is similar to the SSASH study whereby young females in the general population were also more likely to identify as bisexual (C Fisher, personal communication September 2020). Thirty-one per cent of respondents reported they were in a current relationship, either dating or living with their partner. Among 14–15 year old and 16–17 year old age groups, 27% and 34% respectively, were in a relationship at the time of the interview.

Of the total MeH-JOSH sample, 76% (n = 348) reported having ever had oral, vaginal and/or anal sex, which was higher than among school aged young people in the SSASH (57%) and YMM (24%) surveys (Fig 1A and Table 2), but lower than the 97% who reported having had oral, vaginal and/or anal sex in the NSW Young People in Custody Health Survey (YPiCHS) (Fig 1A) [3]. In the MeH-JOSH survey, 74% (n = 338) had had vaginal and/or anal sex compared to 45% in the SSASH survey (Fig 1B and Table 2). Condom use at the last sexual encounter was 55% for both MeH-JOSH and SSASH respondents compared to 64% among YMM respondents (Table 3).

Of the 348 who reported oral, vaginal and/or anal sex, 76% reported having had sex for the first time aged 14 years or less (median age 14 years) and mostly (94%) with a person <18 years of age (median age 15) (Table 2). Justice-involved young females were more likely to have had sex with a person 18 years or older for their first sexual partner (13% versus 3%, $\chi^2(1) = 12.056$, $p = .001$), and their last sexual partner (32% versus 12%, $\chi^2(1) = 19.812$, $p < .001$), compared to males. Of those in the MeH-JOSH survey who had engaged in sex, 60% reported having had oral sex (median age 14 years), 96% vaginal sex (median age 14 years), and 20% anal sex (median age 15 years). Of those who had had sex, reports of oral sexual experience among justice-involved young people (MeH-JOSH) were below the proportions reported by age-matched secondary school students (SSASH) (60% versus 97% respectively), but much higher for vaginal sexual experience (96% versus 76% respectively) (Table 3).

Peak age for vaginal sexual initiation occurred two years earlier in the MeH-JOSH sample compared to the school sample (SSASH) (Fig 2A). Anal sex in those who were sexually active in the justice-involved sample was similar to a national survey on secondary school students (SSASH) (20% versus 22% respectively) although peak age of anal sex initiation occurred about one year earlier among MeH-JOSH participants (Fig 2B).

Of those who have had sex, 41% of justice-involved MeH-JOSH participants reported having six or more sexual partners in their lifetime compared to 16% among school aged young people in the YMM survey, with reported higher numbers of sexual partners at all ages among MeH-JOSH participants (Fig 3A and 3B). Justice-involved young males were more likely to report numbers of 6 or more sexual partners than females (45% versus 34%, $\chi^2(1) = 4.091$, $p = .043$).

For their last sexual encounter, justice-involved young females were more likely to have sex with their current girlfriend/boyfriend (54% versus 40%), while males were more likely to have sex with a stranger (16% versus 4%, $\chi^2(3) = 13.629$, $p = .003$). Young males were more likely to use a condom during their last sexual encounter than females (59% versus 48%, $\chi^2(1) = 4.094$, $p = .043$).

Of the 348 sexually experienced participants, 37% (n = 129) reported prior testing for sexually transmissible infections (STIs) with young females more likely to have been tested recently

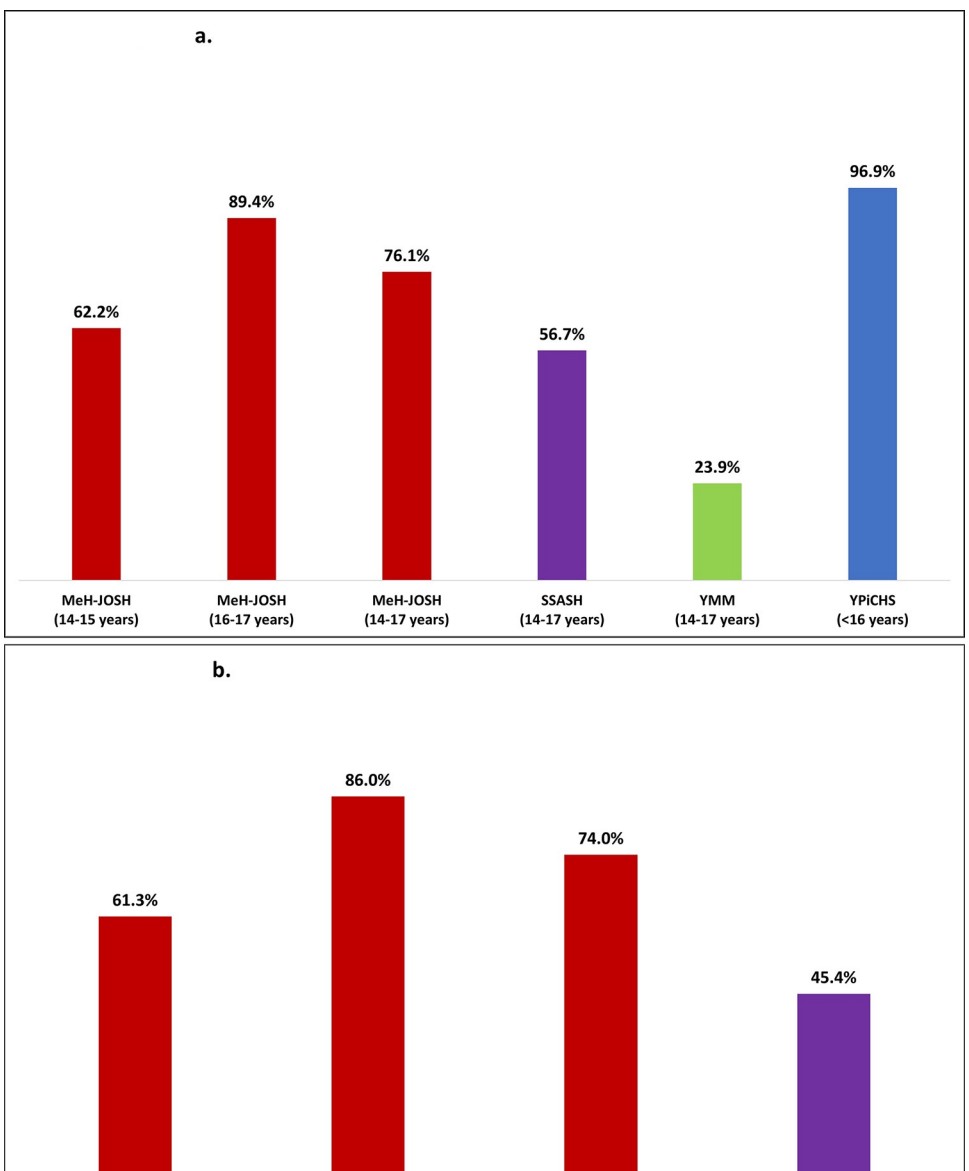

**Fig 1. a.** Proportion of justice-involved and community (SSASH and YMM) young people who had ever had oral, anal and/or vaginal sex. **b.** Proportion of justice-involved and community (SSASH) young people who had ever had anal and/or vaginal sex.

in the last year compared to males (44% versus 24%, $\chi^2(2)$ = 9.392, $p$ = .009). Young females were more likely to be tested by a general practitioner (57% versus 40%), while males were more likely to be tested in a juvenile detention centre or prison (20% versus 3%, $\chi^2(5)$ = 19.377, $p$ = .002). Of those who have ever been tested, 18% were diagnosed with an STI: chlamydia was the most common STI (12.4% positive); none reported being diagnosed with gonorrhoea or syphilis (Table 4).

**Table 2. A comparison of the sexual identity, sexual attraction, sexual behaviours and condom use of community-based justice involved young people (MeH-JOSH) and young people from selected community-based population surveys in Australia, aged between 14 and 17 years.**

| DESCRIPTION | | | Justice-involved population | | | | General population | |
|---|---|---|---|---|---|---|---|---|
| | | | Community based | | | | | |
| | | | MeH-JOSH | | | | 2014 YMM[1] | 2018 SSASH[2] |
| | | | Male | Female | Total | *p* value | Total | Total |
| | | | n (%) | n (%) | n (%) | | n (%) | n (%) |
| | | | | | 465 | | 2004 | 7170 |
| **Sexual identity** | | Heterosexual or straight | 279 (96.2) | 137 (82.5) | 416 (91.2) | <0.001 | - | 5177 (73.6) |
| | | Bisexual | 4 (1.4) | 25 (15.1) | 29 (6.4) | | - | 1147 (16.3) |
| | | Homosexual/Queer | 3 (1.0) | 3 (1.8) | 6 (1.3) | | - | 335 (4.8) |
| | | Unsure/Undecided | 4 (1.4) | 1 (0.6) | 5 (1.1) | | - | 377 (5.4) |
| | | Prefer not to say/Missing* | 3 (-) | 6 (-) | 9 (-) | | - | 134 (-) |
| | | **Total** | **290 (100.0)** | **166 (100.0)** | **456 (100.0)** | | - | **7170 (100)** |
| **Sexual attraction** | | Only to people of the opposite sex | 263 (90.4) | 130 (77.8) | 393 (85.8) | <0.001 | - | 4244 (62.7) |
| | | To both sexes | 9 (3.1) | 23 (13.8) | 32 (7.0) | | - | 2290** (33.8) |
| | | Only to people of own sex | 12 (4.1) | 10 (6.0) | 22 (4.8) | | - | 235 (3.5) |
| | | Not sure | 7 (2.4) | 4 (2.4) | 11 (2.4) | | - | 0 (0.0) |
| | | Prefer not to say/Missing* | 2 (-) | 5 (-) | 7 (-) | | - | 401 (-) |
| | | **Total** | **291 (100.0)** | **167 (100.0)** | **458 (100.0)** | | - | **7170 (100.0)** |
| **Relationships** | | Single/Never married | 211 (72.3) | 109 (63.4) | 320 (69.0) | 0.083 | - | - |
| | | Living with partner in same residence | 14 (4.8) | 15 (8.7) | 29 (6.3) | | - | - |
| | | In a relationship but not living with partner | 67 (22.9) | 48 (27.9) | 115 (24.8) | | - | - |
| | | Prefer not to say/Missing* | 1 (-) | 0 (-) | 1 (-) | | - | - |
| | | **Total** | **292 (100.0)** | **172 (100.0)** | **464 (100.0)** | | - | - |
| **Ever had sex** | **Oral, Vaginal, and/or Anal Sex[3]** | Never had sex | 63 (21.6) | 46 (27.7) | 109 (23.9) | 0.144 | 1411 (76.1) | 3034 (43.3) |
| | | Yes | 228 (78.4) | 120 (72.3) | 348 (76.1) | | 442 (23.9) | 3979 (56.7) |
| | | Prefer not to say/Missing* | 2 (-) | 6 (-) | 8 (-) | | 151 (-) | 157 (-) |
| | | **Total** | **291 (100.0)** | **166 (100.0)** | **457 (100.0)** | | **1853 (100.0)** | **7170 (100.0)** |
| | **Vaginal and/or Anal Sex** | Never had sex | 63 (21.6) | 46 (27.7) | 109 (23.9) | 0.215 | - | - |
| | | No | 8 (2.8) | 2 (1.2) | 10 (2.2) | | - | 3831 (54.6) |
| | | Yes | 220 (75.6) | 118 (71.1) | 338 (73.9) | | - | 3182 (45.4) |
| | | Prefer not to say/Missing* | 2 (-) | 6 (-) | 8 (-) | | - | 157 (-) |
| | | **Total** | **291 (100.0)** | **166 (100.0)** | **457 (100.0)** | | - | **7170 (100.0)** |

[1] 2014 YMM– 2014 Young Minds Matter: 2nd Survey of the Mental Health of Australian Children Survey

[2] 2018 SSASH– 6th National Survey of Secondary Students and Sexual Health

[3] MEH-JOSH–"Can you tell me if you have ever had sex?"; YMM and SSASH–"Have you ever had sexual intercourse?"

* Prefer not to say/Missing not included in the total n (%) and *p*-value calculations

**SSASH–Mostly same, mostly opposite, and equally both sexes combined for "Both Sexes"

Of the total sample, 30% of young males and females recalled being vaccinated against the human papilloma virus (HPV), 44% were not, and 25% had never heard of it or could not recall if they had ever been vaccinated. Young females were more likely to have been HPV

**Table 3.  A comparison of sexual initiation, lifetime sexual encounters and sexual behaviours of community-based justice involved young people (MeH-JOSH) and young people from selected community-based population surveys in Australia, aged between 14 and 17 years.**

| EVER HAD SEX (ORAL, VAGINAL, ANAL SEX) | | | Justice-involved population | | | | General population | |
|---|---|---|---|---|---|---|---|---|
| | | | Community based | | | | 2014 YMM[1] | 2018 SSASH[2] |
| | | | MeH-JOSH | | | | | |
| | | | Male | Female | Total | *p* value | Total | Total |
| | | | n (%) | n (%) | n (%) | | n (%) | n (%) |
| | | | | | 348 | | 442 | 3979 |
| **Sexual initiation** | **Age at sexual initiation** | Total | **226** | **120** | **346** | - | - | - |
| | | *Range* | *6–17* | *5–16* | *5–17* | | - | - |
| | | *Mean* | *13.51* | *13.54* | *13.52* | | - | - |
| | | *Median* | *14* | *14* | *14* | | - | - |
| | **Age of first sexual partner** | <18 years | 218 (96.9) | 102 (87.2) | 320 (93.6) | **0.001** | - | - |
| | | 18 years and over | 7 (3.1) | 15 (12.8) | 22 (6.4) | | - | - |
| | | Prefer not to say/Missing* | 3 (-) | 3 (-) | 6 (-) | | - | - |
| | | **Total** | **225 (100.0)** | **117 (100.0)** | **342 (100.0)** | | - | - |
| | | *Range* | *5–48* | *11–55* | *5–55* | | - | - |
| | | *Mean* | *14.72* | *16.19* | *15.22* | | - | - |
| | | *Median* | *15* | *16* | *15* | | - | - |
| **Lifetime sexual encounters** | **Number of sexual partners[3]** | 1 to 5 people | 124 (55.1) | 79 (66.4) | 203 (59.0) | **0.043** | 371 (83.9) | - |
| | | 6 or more people | 101 (44.9) | 40 (33.6) | 141 (41.0) | | 71 (16.1) | - |
| | | Prefer not to say/Missing* | 3 (-) | 1 (-) | 4 (-) | | 0 (-) | - |
| | | **Total** | **225 (100.0)** | **119 (100.0)** | **344 (100.0)** | | **442 (100.0)** | - |
| | **How often was a condom used?** | All the time | 98 (43.6) | 39 (33.1) | 137 (39.9) | **0.117** | - | - |
| | | Most/Half /Some | 86 (38.2) | 49 (41.5) | 135 (39.4) | | - | - |
| | | Never | 41 (18.2) | 30 (25.4) | 71 (20.7) | | - | - |
| | | Prefer not to say/Missing* | 3 (-) | 2 (-) | 5 (-) | | - | - |
| | | **Total** | **225 (100.0)** | **118 (100.0)** | **343 (100.0)** | | - | - |
| **Last sexual encounter** | **Was the last person you had sex with . . .?** | •Your current girlfriend/boyfriend | 91 (40.1) | 65 (54.2) | 156 (45.0) | **0.003** | - | 2079 (64.3) |
| | | • Someone you had known for a while, but had not had sex with before | 54 (23.8) | 29 (24.2) | 83 (23.9) | | - | 574 (17.7) |
| | | • Someone known for a while, had sex with before, but not your current girlfriend or boyfriend | 44 (19.4) | 21 (17.5) | 65 (18.7) | | - | 370 (11.4) |
| | | • Someone you had just met for the first time | 38 (16.7) | 5 (4.2) | 43 (12.4) | | - | 213 (6.6) |
| | | • Prefer not to say/Missing | 1 (-) | 0 (-) | 1 (-) | | - | 743 (-) |
| | | **Total** | **227 (100.0)** | **120 (100.0)** | **347 (100.0)** | | - | **3979 (100.0)** |

(*Continued*)

**Table 3.** (Continued)

| EVER HAD SEX (ORAL, VAGINAL, ANAL SEX) | | | Justice-involved population | | | | General population | |
|---|---|---|---|---|---|---|---|---|
| | | | Community based | | | | | |
| | | | MeH-JOSH | | | | 2014 YMM[1] | 2018 SSASH[2] |
| | | | Male | Female | Total | p value | Total | Total |
| | | | n (%) | n (%) | n (%) | | n (%) | n (%) |
| | | | | | 348 | | 442 | 3979 |
| | Age of last sexual partner | <18 years | 196 (87.9) | 81 (68.1) | 277 (81.0) | <0.001 | - | 2449 (76.6) |
| | | 18 years and over | 27 (12.1) | 38 (31.9) | 65 (19.0) | | - | 749 (23.4) |
| | | Prefer not to say/Missing* | 5 (-) | 1 (-) | 6 (-) | | - | 141 (-) |
| | | Total | 223 (100.0) | 119 (100.0) | 342 (100.0) | | - | 3198 (100.0) |
| | | Range | 11–48 | 12–35 | 11–48 | | - | - |
| | | Mean | 16.29 | 17.35 | 16.66 | | - | - |
| | | Median | 16 | 17 | 16 | | - | - |
| | Did you use a condom? | No | 92 (40.7) | 62 (52.1) | 154 (44.6) | 0.043 | 154 (34.8) | 1398 (43.0) |
| | | Yes | 134 (59.3) | 57 (47.9) | 191 (55.4) | | 284 (64.3) | 1807 (55.6) |
| | | Not sure | 0 (0.0) | 0 (0.0) | 0 (0.0) | | 4 (0.9) | 47 (1.4) |
| | | Prefer not to say/Missing* | 2 (-) | 1 (-) | 3 (-) | | 0 (-) | 87 (-) |
| | | Total | 226 (100.0) | 119 (100.0) | 345 (100.0) | | 442 (100.0) | 3252 (100.0) |
| Oral sex | Ever had oral sex | No | 84 (38.4) | 53 (44.5) | 137 (40.5) | 0.269 | - | 119 (3.0) |
| | | Yes | 135 (61.6) | 66 (55.5) | 201 (59.5) | | - | 3860 (97.0) |
| | | Prefer not to say/Missing* | 9 (-) | 1 (-) | 10 (-) | | - | 0 (-) |
| | | Total | 219 (100.0) | 119 (100.0) | 338 (100.0) | | - | 3979 (100.0) |
| | Age at sexual initiation (oral sex) | Total | 135 | 65 | 200 | - | - | - |
| | | Range | 9–17 | 5–17 | 5–17 | | - | - |
| | | Mean | 14.06 | 13.97 | 14.03 | | - | - |
| | | Median | 14 | 14 | 14 | | - | - |
| Vaginal sex | Ever had vaginal sex | No | 10 (4.4) | 3 (2.5) | 13 (3.8) | 0.366 | - | 807 (21.1) |
| | | Yes | 215 (95.6) | 117 (97.5) | 332 (96.2) | | - | 3022 (78.9) |
| | | Prefer not to say/Missing* | 3 (-) | 0 (-) | 3 (-) | | - | 150 (-) |
| | | Total | 225 (100.0) | 120 (100.0) | 345 (100.0) | | - | 3979 (100.0) |
| | Age at sexual initiation (vaginal sex) | Total | 212 | 117 | 329 | - | - | - |
| | | Range | 6–17 | 5–16 | 5–17 | | - | - |
| | | Mean | 13.61 | 13.71 | 13.64 | | - | - |
| | | Median | 14 | 14 | 14 | | - | - |

(Continued)

**Table 3.** (Continued)

| EVER HAD SEX (ORAL, VAGINAL, ANAL SEX) | | | Justice-involved population | | | | General population | |
|---|---|---|---|---|---|---|---|---|
| | | | Community based | | | | | |
| | | | MeH-JOSH | | | | 2014 YMM[1] | 2018 SSASH[2] |
| | | | Male | Female | Total | p value | Total | Total |
| | | | n (%) | n (%) | n (%) | | n (%) | n (%) |
| | | | | | 348 | | 442 | 3979 |
| | How often was a condom used? | All the time | 95 (44.4) | 40 (34.5) | 135 (40.9) | 0.151 | - | - |
| | | Most/Half/Some | 85 (39.7) | 50 (43.1) | 135 (40.9) | | - | - |
| | | Never | 34 (15.9) | 26 (22.4) | 60 (18.2) | | - | - |
| | | Prefer not to say/Missing* | 1 (-) | 1 (-) | 2 (-) | | - | - |
| | | **Total** | **214 (100.0)** | **116 (100.0)** | **330 (100.0)** | | - | - |
| Anal sex | Ever had anal sex | No | 180 (79.3) | 99 (82.5) | 279 (80.4) | 0.474 | - | 3114 (78.3) |
| | | Yes | 47 (20.7) | 21 (17.5) | 68 (19.6) | | - | 865 (21.7) |
| | | Prefer not to say/Missing* | 1 (-) | 0 (-) | 1 (-) | | - | 0 (-) |
| | | **Total** | **227 (100.0)** | **120 (100.0)** | **347 (100.0)** | | - | **3979 (100.0)** |
| | **Age at sexual initiation (anal sex)** | **Total** | **47** | **21** | **68** | - | - | - |
| | | *Range* | *9–17* | *12–16* | *9–17* | | - | - |
| | | *Mean* | *14.57* | *14.81* | *14.65* | | - | - |
| | | *Median* | *15* | *15* | *15* | | - | - |
| | How often was a condom used? | All the time | 25 (53.2) | 6 (30.0) | 31 (46.3) | 0.218 | - | - |
| | | Most/Half/Some | 6 (12.8) | 4 (20.0) | 10 (14.9) | | - | - |
| | | Never | 16 (34.0) | 10 (50.0) | 26 (38.8) | | - | - |
| | | Prefer not to say/Missing* | 0 (-) | 1 (-) | 1 (-) | | - | - |
| | | **Total** | **47 (100.0)** | **20 (100.0)** | **67 (100.0)** | | - | - |

[1] 2014 YMM– 2014 Young Minds Matter: 2nd Survey of the Mental Health of Australian Children Survey

[2] 2018 SSASH– 6th National Survey of Secondary Students and Sexual Health

[3] MEH-JOSH–"During your life, with how many people have you had sex?"; YMM–"During your life, with how many people have you had sexual intercourse?"

* Prefer not to say/Missing not included in the total n (%) and p-value calculations

vaccinated than males (38% versus 26%, $\chi^2(2) = 7.361$, $p = .025$). Of those who reported being vaccinated (n = 139), Aboriginal and/or Torres Strait Islanders had lower vaccination rates compared to non-Indigenous (37% versus 63%, $\chi^2(2) = 6.596$, $p = .037$). Among those who were never vaccinated or were unsure or had never heard of HPV, 58% indicated they were willing to have the HPV vaccine (Table 5). A higher proportion of young females than males indicated a willingness to be HPV vaccinated (68% versus 53%, $\chi^2(2) = 6.988$, $p = .030$).

## Discussion

We have observed that young people involved in the justice system have a higher engagement in sex compared with their peers in school surveys (1.3 to 3.2 times higher); (ii) are often starting to have sex at a younger age (median age 14 years compared to 17 years in the general Australian population and 15 years in the general Indigenous population) [17, 18]; (iii) are

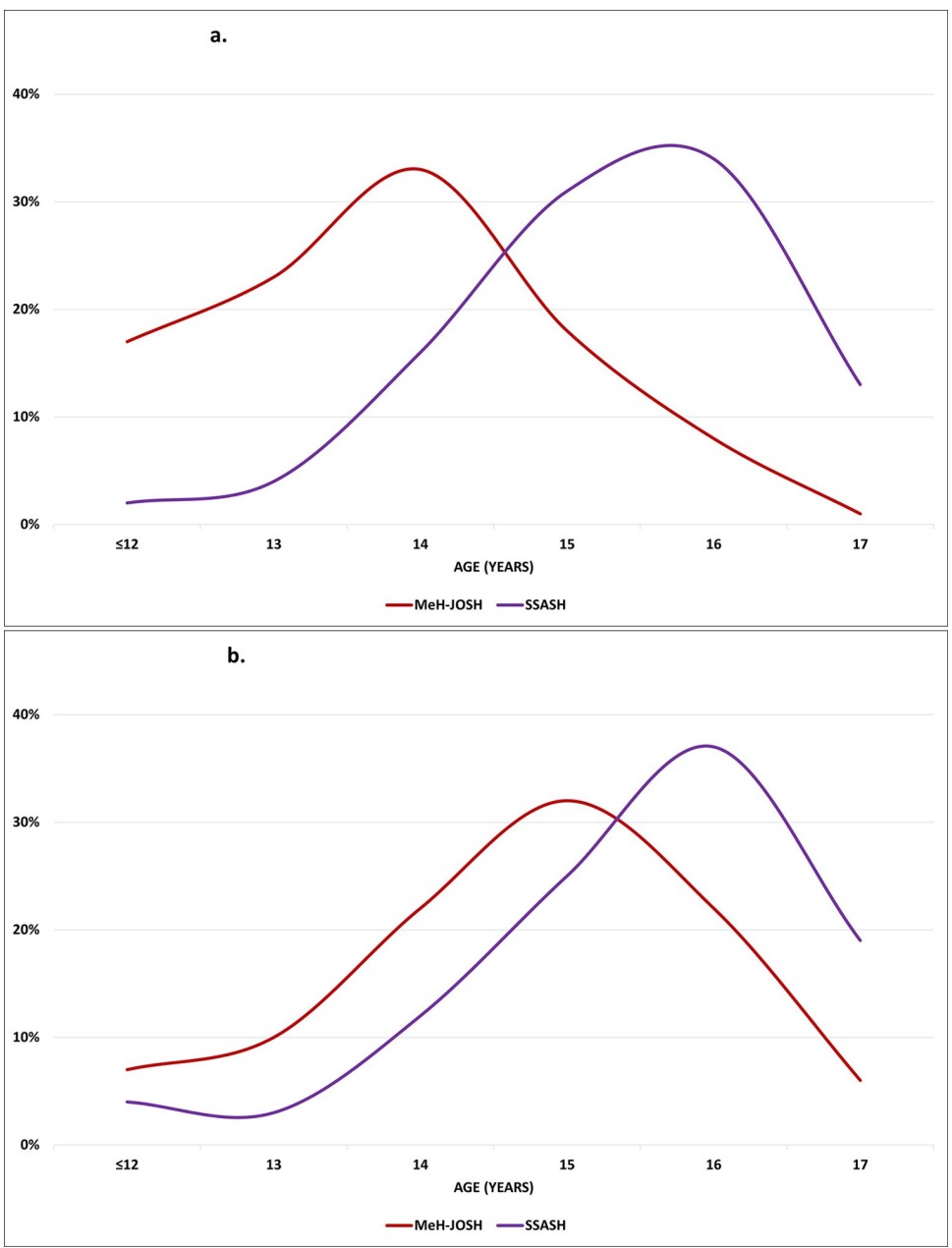

**Fig 2. a.** Age of sexual initiation (vaginal) among justice-involved and community (SSASH) young people. **b.** Age of sexual initiation (anal) among justice-involved and community (SSASH) young people.

significantly more likely to report large numbers of sexual partners (6+) at a young age. While similar rates of young people used a condom during their last sexual encounter across MeH-JOSH, SSASH and YMM surveys, engagement in more frequent and riskier sexual behaviours suggests that this may pose more of a STI risk to justice-involved young people. Most had never been or could not recall having been vaccinated for HPV, even though all Australian young people are offered the vaccine in school at age 12–13 years and first-dose coverage in young females is 86% and 78% in young males, and increasing [19]. These findings highlight the critical need for sexual health promotion and vaccination programs to reach this

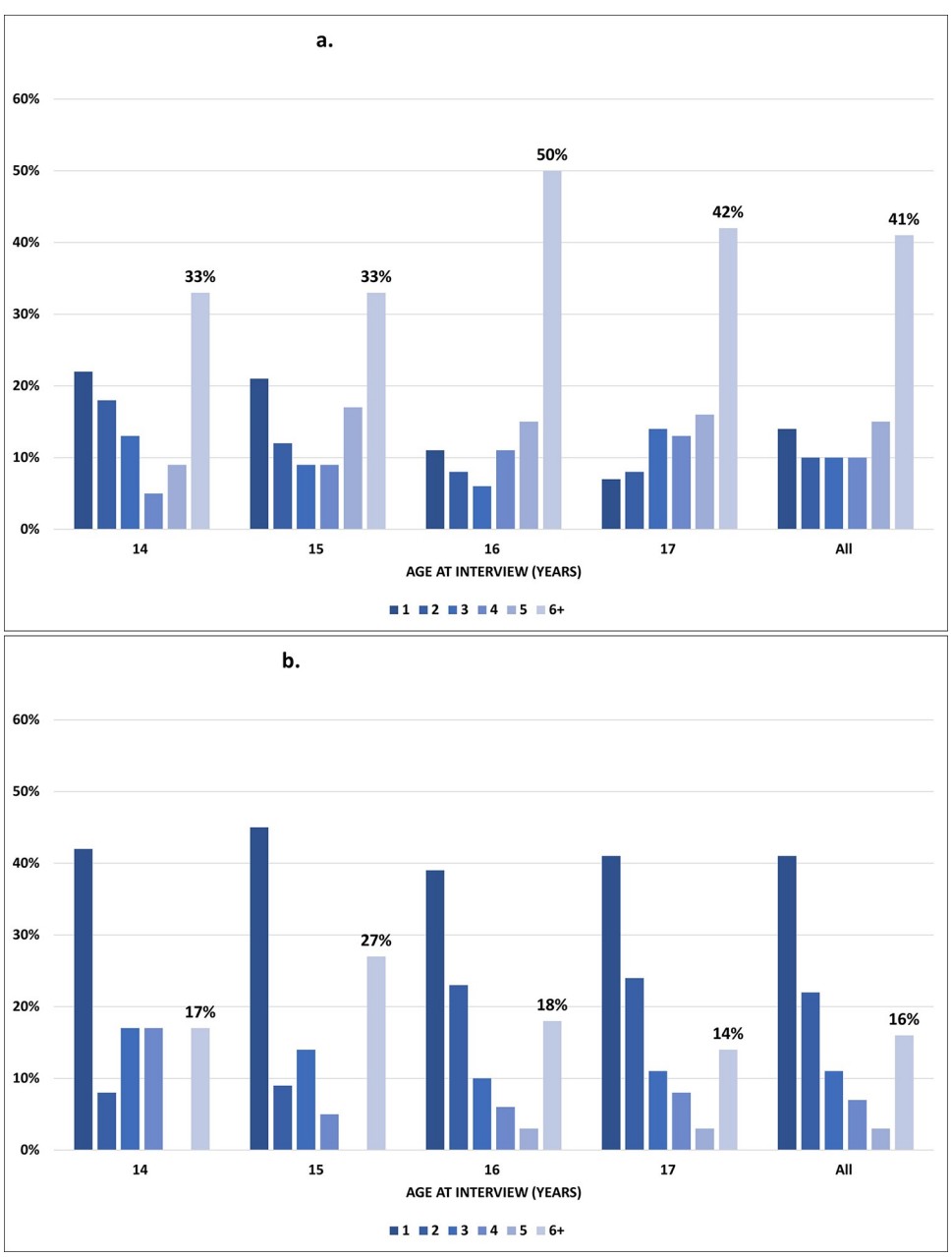

**Fig 3. a.** Proportion of sexual partners by age in justice-involved young people. *MeH-JOSH–Of those who had ever had oral, vaginal and/or anal sex (n = 344).* **b.** Proportion of sexual partners by age in community (YMM) young people. *YMM–Of those who had ever had sexual intercourse (n = 442).*

population, particularly those of Aboriginal and/or Torres Strait Islander descent. It also represents an opportunity; 58% of participants in this survey (who had never been or were unsure if they had been vaccinated) indicated their willingness to be vaccinated for HPV.

Young people who identified as Aboriginal and/or Torres Strait Islander (Indigenous) descent were overrepresented in the survey sample compared with the general population (44% versus 5%), reflecting the greater involvement of this population in the justice system. In 2016–2017, on an average day, there were 4,286 young people between the age of 14 and 17

**Table 4. STI testing and diagnosis as reported by community-based justice involved young people (MeH-JOSH) in Australia, aged between 14 and 17 years.**

| EVER HAD SEX (ORAL, VAGINAL, ANAL SEX) | | Justice-involved population | | | |
|---|---|---|---|---|---|
| | | Community based | | | |
| | | MeH-JOSH | | | |
| | | Male | Female | Total | *p* value |
| | | n (%) | n (%) | n (%) | |
| | | 228 | 120 | 348 | |
| **Ever tested for an STI/HIV/AIDS** | No | 157 (69.2) | 58 (49.2) | 215 (62.3) | **0.009** |
| | Yes, in the last year | 55 (24.2) | 52 (44.1) | 107 (31.0) | |
| | Yes, more than a year ago | 15 (6.6) | 8 (6.8) | 23 (6.7) | |
| | Prefer not to say/Missing* | 1 (-) | 2 (-) | 3 (-) | |
| | **Total** | **227 (100.0)** | **118 (100.0)** | **345 (100.0)** | |
| *If YES, where did you get your last STI and/or HIV test?* | Local doctor at a General Practice clinic | 28 (40.0) | 34 (56.7) | 62 (47.7) | **0.002** |
| | Juvenile Justice Detention/Prison | 14 (20.0) | 2 (3.3) | 16 (12.3) | |
| | Hospital | 8 (11.4) | 4 (6.7) | 12 (9.2) | |
| | Family planning/Sexual health clinic | 2 (2.9) | 9 (15.0) | 11 (8.5) | |
| | Aboriginal Medical Service | 4 (5.7) | 6 (10.0) | 10 (7.7) | |
| | Other | 14 (20.0) | 5 (8.3) | 19 (14.6) | |
| | **Total** | **70 (100.0)** | **60 (100.0)** | **130 (100.0)** | |
| **(If ever tested) Ever diagnosed with an STI/HIV/AIDS** | No | 60 (87.0) | 46 (76.7) | 106 (82.2) | **0.312** |
| | Yes, in the last year | 6 (8.7) | 9 (15.0) | 15 (11.6) | |
| | Yes, more than a year ago | 3 (4.3) | 5 (8.3) | 8 (6.2) | |
| | Prefer not to say/Missing* | 1 (-) | 0 (-) | 1 (-) | |
| | **Total** | **69 (100.0)** | **60 (100.0)** | **129 (100.0)** | |
| *If YES, which infections have you been diagnosed with? (May have more than one answer)* | Chlamydia | 7 (87.5) | 9 (81.8) | 16 (84.2) | **0.652** |
| | Trichomoniasis | 1 (12.5) | 1 (9.1) | 2 (10.5) | |
| | Urinary Tract Infection | 0 (0.0) | 1 (9.1) | 1 (5.3) | |
| | Genital warts | 0 (0.0) | 1 (9.1) | 1 (5.3) | |
| | Prefer not to say/Missing* | 1 (-) | 3 (-) | 4 (-) | |
| | **Total** | **8 (100.0)** | **12 (100.0)** | **20 (100.0)** | |

*Prefer not to say/Missing not included in the total n (%) and *p*-value calculations

years under youth or juvenile justice supervision in Australia, of whom 2,036 (48%) were Aboriginal and/or Torres Strait Islander [20].

Of the 129 survey participants who reported ever being tested for an STI, 18% had tested positive in the past, with most being diagnosed with chlamydia. In Australia, chlamydia infections are the most frequently notified STI and are increasing with females 2.3 times more likely than males to be diagnosed each year, while diagnosis in Aboriginal and/or Torres Strait Islander populations is 2.8 times higher than non-Indigenous populations (1193.9 per 100,000 compared with 427.0 per 100,000) [21]. In 2017, 17% of 100,775 notifications were among those aged 15–19 years underscoring the need to reach these justice involved young people who fall within this age group and are at sexual health risk [21].

A limitation of this study includes the use of a non-random sampling strategy (necessary to overcome practical difficulties of reaching such individuals), resulting in a sample that may not be statistically representative of the target population (i.e. the full range of justice-involved

**Table 5. HPV vaccinations as reported by community-based justice involved young people in Australia, aged between 14 and 17 years.**

| DESCRIPTION | | Justice-involved population | | | |
| --- | --- | --- | --- | --- | --- |
| | | Community based | | | |
| | | MeH-JOSH | | | |
| | | Male | Female | Total | p value |
| | | n (%) | n (%) | n (%) | |
| | | 293 | 172 | 465 | |
| **Have you ever been vaccinated for HPV?** | Yes* | 76 (26.2) | 63 (38.4) | 139 (30.6) | **0.025** |
| | No | 137 (47.2) | 64 (39.0) | 201 (44.3) | |
| | Never heard of it /Don't know/Unsure | 77 (26.6) | 37 (22.6) | 114 (25.1) | |
| | Prefer not to say/ Missing** | 3 (-) | 8 (-) | 11 (-) | |
| | **Total** | **290 (100.0)** | **164 (100.0)** | **454 (100.0)** | |
| *If YES, how many doses of HPV vaccine did you receive?* | 1 Dose | 11 (14.7) | 11 (17.5) | 22 (15.9) | **0.764** |
| | 2 Doses | 14 (18.7) | 13 (20.6) | 27 (19.6) | |
| | 3 Doses | 10 (13.3) | 5 (7.9) | 15 (10.9) | |
| | Don't know/Unsure | 40 (53.3) | 34 (54.0) | 74 (53.6) | |
| | Prefer not to say/ Missing** | 1 (-) | 0 (-) | 1 (-) | |
| | **Total** | **75 (100.0)** | **63 (100.0)** | **138 (100.0)** | |
| *If NO/NEVER HEARD OF IT/DON'T KNOW/UNSURE, do you want the HPV vaccine to protect you against [FEMALE: cervical cancer] [MALE: cancers of the anus, mouth/throat, or penis]?* | No | 83 (38.8) | 25 (24.8) | 108 (34.3) | **0.030** |
| | Yes | 113 (52.8) | 69 (68.3) | 182 (57.8) | |
| | Don't know / Unsure | 18 (8.4) | 7 (6.9) | 25 (7.9) | |
| | **Total** | **214 (100.0)** | **101 (100.0)** | **315 (100.0)** | |

*Of 139 who reported HPV vaccination, Aboriginal and/or Torres Strait Islander (Indigenous) 37% (n = 51), Non-Indigenous 63% (n = 87)

**Prefer not to say/Missing not included in the total n (%) and *p*-value calculations

young people). The study data were compared with a sample of similarly aged secondary students (SSASH) [16] and interpretations of the results should take into account the socio-demographic differences between the two study populations (e.g. education and Aboriginality). Sexual behaviours were also contrasted from YMM (2014) [15] and the Young People in Custody Health Survey in New South Wales (2015) [3] to situate the data with other studies in the general population and in juvenile detention. Another limitation of this study is that it was not possible to conduct formal comparisons across these different survey samples.

Findings are based on self-report and may be impacted by recall or social desirability bias. While memory recall is not altogether an accurate indicator other studies in Australia also reveal similar results for similar questions on HPV (e.g. SSASH 2018), most do not recall if they had been vaccinated for HPV which is an interesting finding in itself. Perhaps in future studies we can ask respondents for their consent to match them up with their vaccination records.

Since this was an exploratory study, we did not hypothesise about differences between the incarcerated youth justice population and those with less involvement. Differences stratified by involvement with the justice system can be explored in future.

This research adds to the dearth of surveys of justice involved young people in the community [4, 22]. The study demonstrates how young people involved in the justice system may be at high risk of STIs and that it is possible to target this usually hard to access population through youth oriented community spaces without juvenile justice or corrective services involvement. Permission for access and research from these government departments is sometimes an arduous or impossible process despite the urgent need to address serious health issues in this population. Reaching justice involved young people in the community presents another viable alternative to enhance the delivery of sexual health promotion programmes and services and creates an opportunity to augment it with existing health services (mental health, alcohol and substance abuse) currently provided in these youth oriented community and drop-in centres.

## Supporting information

**S1 File.**
(DOCX)

## Acknowledgments

We would like to acknowledge the participation and assistance by the Western Australia Department of Justice, Western Australia Department of Health, Queensland and Western Australia community youth based organisations and drop-in centres, flexi-schools and colleges, and young people who agreed to be part of the survey as a whole in the conduct of this research.

## Author Contributions

**Conceptualization:** Lorraine Yap.

**Data curation:** Lorraine Yap.

**Formal analysis:** Lorraine Yap, Megan Steele, Christopher Fisher, Bianca Ton.

**Funding acquisition:** Lorraine Yap, Basil Donovan, Sally Nathan, Elizabeth Sullivan, Tony Butler.

**Investigation:** Lorraine Yap, Jocelyn Jones, Basil Donovan, Sally Nathan, Sophie Davison, Alun Richards, Tony Butler.

**Methodology:** Lorraine Yap, Basil Donovan.

**Project administration:** Lorraine Yap, Jocelyn Jones, Ed Heffernan.

**Resources:** Ed Heffernan.

**Supervision:** Lorraine Yap.

**Visualization:** Lorraine Yap, Bianca Ton.

**Writing – original draft:** Lorraine Yap.

**Writing – review & editing:** Lorraine Yap, Jocelyn Jones, Basil Donovan, Sally Nathan, Elizabeth Sullivan, Sophie Davison, Ed Heffernan, Alun Richards, Carla Meurk, Megan Steele, Christopher Fisher, Bianca Ton, Tony Butler.

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
