## [Decision Letter · Decision Letter 0]

2 Jan 2020

PONE-D-19-26106

The sexual behaviours of adolescents  aged between 14 and 17 years involved with the juvenile justice system in Australia: a community-based survey

PLOS ONE

Dear Dr Yap,

Thank you for submitting your manuscript to PLOS ONE. After careful consideration, we feel that it has merit but does not fully meet PLOS ONE’s publication criteria as it currently stands. Therefore, we invite you to submit a revised version of the manuscript that addresses the points raised during the review process.

We would appreciate receiving your revised manuscript by Feb 16 2020 11:59PM. To enhance the reproducibility of your results, we recommend that if applicable you deposit your laboratory protocols in protocols.io, where a protocol can be assigned its own identifier (DOI) such that it can be cited independently in the future. For instructions see: http://journals.plos.org/plosone/s/submission-guidelines#loc-laboratory-protocols

We look forward to receiving your revised manuscript.

Kind regards,

Andrea Knittel

Academic Editor

PLOS ONE

2. You indicated that you had ethical approval for your study. In your Methods section, please ensure you have also stated whether you obtained consent from parents or guardians of the minors included in the study or whether the research ethics committee or IRB specifically waived the need for their consent. In this manuscript parental consent is only referred to for those minors recruited from the Children's Court. Additionally, minors were recruited from "youth mental health service centres". Please confirm that this vulnerable population were determined to be capable of providing consent. Furthermore, the site of recruitment was not included as a potentially confounding factor in the statistical analyses. Please justify why this was not performed.

3. Please include additional information regarding the survey or questionnaire used in the study and ensure that you have provided sufficient details that others could replicate the analyses. For instance, if you developed a questionnaire as part of this study and it is not under a copyright more restrictive than CC-BY, please include a copy, in both the original language and English, as Supporting Information. Additionally, please refer to any post-hoc corrections made during your statistical analysis for multiple comparisons. Please justify the reasons if these were not performed.

Additional Editor Comments (if provided):

The reviewers identified several ways to make the paper clearer to the reader. I am in agreement with both reviewers that this is a useful study that will likely ultimately merit publication, but the authors will need to accomplish all of the revisions identified below in order to meet the publication criteria.

Reviewers' comments:

Reviewer's Responses to Questions

**Comments to the Author**

1. Is the manuscript technically sound, and do the data support the conclusions?

Reviewer #1: Yes

Reviewer #2: No

2. Has the statistical analysis been performed appropriately and rigorously? 

Reviewer #1: Yes

Reviewer #2: No

3. Have the authors made all data underlying the findings in their manuscript fully available?

Reviewer #1: Yes

Reviewer #2: Yes

4. Is the manuscript presented in an intelligible fashion and written in standard English?

Reviewer #1: Yes

Reviewer #2: No

5. Review Comments to the Author

Reviewer #1: Overall, this is a useful descriptive study of reproductive health needs.

Abstract

States higher risk of STI. And pregnancy?

Intro

Can authors site what is known about repro risk in JJ youth? Also, any hypothesized differences between kids with low levels of involvement vs longer-term incarceration

Methods

Do not understand why approach necessary. Can authors clarify? Why were the selection criteria chosen? Whose missing from the sample since its not representative?

Exactly 42 mintues? seems awkward. suggest saying 40 min.

consent process is stated for only 1 of the recruitment methods

Discussion

First sentence very long

Do authors think youth recalling HPV vaccination correctly? Is never been vaccinated variable category based on records or youth recall?

I'm not Austrialian. Familiar with Aboriginal but what's the significant of Torres Strait Islander? Perhaps explain briefly in Intro of Methods for international audience. I'm guessing these are also "indigenous" youth, but not sure.

Regarding disproportionate confinement, what is the prevalence of aboriginal and islanders on the island?

Limitations

Authors' frustration with approval process and resultant sampling framework comes through. perhaps revise

I would like to see the authors expand on differences between kids of varying levels of contact. Did they do sub-analyses? At the very least, Discussion should comment on this. Could be added as a limitation

Table

Suggest title column with name that clearly indicates that MEH group is the JJ group

Formatting of tables could be improved. Suggest n and then % in parenthesis (unless authors are following PLOS table formatting guidelines that I am unaware of). It will make table more digestible.

Footnotes should briefly explain what the different samples are.

What is the rationale for the bolding? It seems inconsistent.

Reviewer #2: This study offered a descriptive comparison of sexual behaviors among justice-involved and community youth in Australia. The authors interviewed adolescents currently or previously involved with the criminal justice system, and compared their responses to non-delinquent peers using alternative data sources. Youth in the justice-involved sample were more likely to engage in sex and start having sex at a younger age, and were more likely to report having 6 or more partners.

Overall, I think the premise of this paper is a useful contribution. While researchers argue that justice-involved youth are at heightened risk of sexual risk-taking, there are few formal comparisons with nationally representative samples to empirically support this assertion. Although this paper provides evidence for these assumptions, there are a number of substantial issues that would need to be addressed prior to publication.

Major issues

1. The authors could improve the introduction by making a stronger argument as to why delinquent populations may be at risk for sexual risk-taking. Right now, it seems as though their argument rests on the idea that delinquent populations may not be reached by traditional intervention efforts, and ignores the theoretical and empirical work that suggests that the same factors that put youth at risk for delinquency may drive their tendency to engage in riskier sexual behaviors. I would recommend reviewing a number of articles:

a. Robbins, R. N., & Bryan, A. (2004). Relationships between future orientation, impulsive sensation seeking, and risk behavior among adjudicated adolescents. Journal of adolescent research, 19(4), 428-445.

b. Bryan, A. D., Schmiege, S. J., & Magnan, R. E. (2012). Marijuana use and risky sexual behavior among high-risk adolescents: trajectories, risk factors, and event-level relationships. Developmental psychology, 48(5), 1429.

c. In general, Angela Bryan has a number of relevant articles on this subject that are worth reading.

2. Similarly, although formal comparisons with community populations are limited, there are a few studies that address a similar subject matter that seem like they should be included:

a. Elkington, K. S., Teplin, L. A., Mericle, A. A., Welty, L. J., Romero, E. G., & Abram, K. M. (2008). HIV/sexually transmitted infection risk behaviors in delinquent youth with psychiatric disorders: A longitudinal study. Journal of the American Academy of Child & Adolescent Psychiatry, 47(8), 901-911.

b. Teplin, L. A., Mericle, A. A., McClelland, G. M., & Abram, K. M. (2003). HIV and AIDS risk behaviors in juvenile detainees: Implications for public health policy. American Journal of Public Health, 93(6), 906-912.

c. Teplin, L. A., Elkington, K. S., McClelland, G. M., Abram, K. M., Mericle, A. A., & Washburn, J. J. (2005). Major mental disorders, substance use disorders, comorbidity, and HIV-AIDS risk behaviors in juvenile detainees. Psychiatric Services, 56(7), 823-828.

d. DiClemente, R. J., Lanier, M. M., Horan, P. F., & Lodico, M. (1991). Comparison of AIDS knowledge, attitudes, and behaviors among incarcerated adolescents and a public school sample in San Francisco. American journalof public health,81(5), 628–630.

3. The authors should provide more information on what they mean by a “purposive sampling design.” The authors do not explain what the calculated sample size was, which is odd given the heading of the section.

4. The authors should offer some information as to if the sample differed across the four different recruitment methods, given that they are ultimately combined.

5. The brevity of the methods makes it difficult to discern how the authors measured sexual behavior. Many of the terms used are quite broad (justice system involvement, sexual history, sexual knowledge) and could encompass a wide variety of topics. The authors should expand on this section and provide more details as to what their measures asked, providing specific examples of the questions.

6. The authors describe the comparison sample under “Data Analysis” but it seems like this information should be included in the methods section, in the same section where the justice-system sample is described. We are also given very few details about these comparison samples and some additional information on these separate studies would be helpful, as well as an explanation of why two different studies were employed. If the two studies included different variables, then the authors should describe the different measures covered in each study. Similarly, there is very little information provided on what method of matching was used. It also seems as though the authors only matched on age, and it is unclear if other characteristics (gender, SES, race/ethnicity) were considered in the matching process. Given the possible differences in the two populations (justice involved vs. nationally representative), it is possible that these differences could explain different rates of sexual risk-taking, beyond justice-system involvement.

7. The results appeared a bit disorganized, and as a result were difficult to follow. First, I think it would be helpful for the authors to build on their analytic plan in the Data Analysis section. Although they note the use of Chi Square tests, it would be helpful to explain which variables they are comparing (and in which samples) before jumping into the results. The first paragraph under “Sexual health and behaviours” seems like it should be earlier in the methods, not in the results. A similar set of information on the two nationally representative samples would also be helpful.

a. The authors report differences in sexual orientation by gender, but it is not clear 1) what sample this is referring to (justice involved, nationally representative, both combined) and 2) why a gender comparison was made. It does not seem like this was part of the goal of the paper, and the authors did not make any comparisons between the different samples. It is evident from Table 2 that this information is just referring to the justice-involved sample, but it is not clear why no formal comparisons were made with the SSASH sample.

b. The next paragraph, “Of the total sample, 76%...” also needs to be clarified. It would be easier to interpret if the authors provided the percentages for both groups, rather than providing a percentage and then stating it was 1.3-3.2x higher than the other groups. I’m also not sure what the next sentence (“However, this was lower than the 92%...) refers to. Based on Figure 1 it looks like these analyses were split by age, but this is not clear from the methods or results. In addition, it is not clear if the original item combined oral, vaginal and/or anal sex, or if this was combined by the authors. It would be preferable to look at these types of sexual behaviors separately, if possible.

c. The next section (“Of the 348 who reported…) also lacks organization. It would be helpful to provide comparable information across samples, and then provide a formal comparison using a statistical test. For example, 76% of youth in the justice-sample reported having sex for the first time at age 14 or younger. What is the comparable statistic for the other sample? The authors instead provide some descriptive information about the justice-sample, but then offer only a select number of comparisons (e.g., how oral sex was less common and vaginal sex was more common among justice-system participants). There is also no formal statistic provided to show whether these differences are statistically significant.

d. The authors indicate that justice-system youth initiated sexual activity at an earlier age, but it is not clear if this was formally tested. What statistical test did they use?

e. For sexual initiation, it is confusing to say 41% of justice-involved participants reported 6+ partners, which was 2.6x higher than the comparison sample. It would be helpful just to provide the percentage for the other samples as well.

f. The last section (“Of the 348 sexually experienced participants, 37%...”), it appears the authors only provide descriptive information for the justice involved sample. Without a formal comparison it is difficult to discern if these percentages are low, high or average.

g. Overall, the results would need to be entirely restructured to offer descriptive statistics for all samples, and then provide formal comparisons using the appropriate statistical tests.

8. The discussion does not entirely map onto the results from the paper. For example, the authors argue justice system youth are at greater risk for STIs, but this ignores the finding that their condom use is comparable to the community samples. The authors also argue that most had not been tested for HPV, but we have no comparison to the community sample to know if this is something that is particular to justice system populations. It seems like the primary finding is that justice system youth are more likely to have had sex, but given that all sex is not inherently risky, it seems like some of these other statistics are worth discussing (condom use, HPV testing, HIV testing).

Minor issues

1. On page 4, it should read, “The Mental Health, Sexual Health and Reproductive…study aimed to describe” (rather than describes).

2. On the top of page 5, “The main aim of the survey was…” (not were).

3. The authors switch back and forth between saying justice involved vs. using the MeH-JOSH acronym. It would be helpful if this was consistent.

4. Table 2 is overwhelming to look at, and could be broken down into smaller pieces for easier interpretation.

6. PLOS authors have the option to publish the peer review history of their article (what does this mean?). If published, this will include your full peer review and any attached files.

Reviewer #1: No

Reviewer #2: No

---

## [Author Response · Author response to Decision Letter 0]

17 Aug 2020

2. You indicated that you had ethical approval for your study. In your Methods section, please ensure you have also stated whether you obtained consent from parents or guardians of the minors included in the study or whether the research ethics committee or IRB specifically waived the need for their consent. In this manuscript parental consent is only referred to for those minors recruited from the Children's Court. Additionally, minors were recruited from "youth mental health service centres". Please confirm that this vulnerable population were determined to be capable of providing consent. 

We have added to the manuscript:

“Due to the nature of this population and from our experiences of conducting surveys among young offenders in Australia[3], we expected that a high proportion of young people would not have a good relationship with their parent(s) or with the adults responsible for them. Thus human research ethics approval was sought and given for young people recruited in the survey to be treated as mature minors.[14] Nevertheless, all young people approached were asked by recruiters if we could contact their parents or guardians for permission to allow them to participate in the survey. Almost all respondents refused us contact or to give any contact details of their parents or guardians preferring to give consent themselves. Recruiters were required to administer a Gillick Competency checklist to ensure that all respondents met the criteria of a mature minor.”

Furthermore, the site of recruitment was not included as a potentially confounding factor in the statistical analyses. Please justify why this was not performed.

Site of recruitment was not a confounding factor for age and gender although it was significant for Aboriginality (p<0.05). 

 3. Please include additional information regarding the survey or questionnaire used in the study and ensure that you have provided sufficient details that others could replicate the analyses. For instance, if you developed a questionnaire as part of this study and it is not under a copyright more restrictive than CC-BY, please include a copy, in both the original language and English, as Supporting Information. Additionally, please refer to any post-hoc corrections made during your statistical analysis for multiple comparisons. Please justify the reasons if these were not performed.

Parts of the questionnaire (MINI KID 6.0) are under copyright restrictions (D. Sheehan). 

This paper focuses on the descriptive analysis of the MeH-JOSH and data obtained from other studies. Multivariate analyses were not performed for the main tables in this paper.

Additional Editor Comments (if provided):

The reviewers identified several ways to make the paper clearer to the reader. I am in agreement with both reviewers that this is a useful study that will likely ultimately merit publication, but the authors will need to accomplish all of the revisions identified below in order to meet the publication criteria.

Reviewers' comments:

Reviewer's Responses to Questions

Comments to the Author

1. Is the manuscript technically sound, and do the data support the conclusions?

Reviewer #1: Yes

Reviewer #2: No

2. Has the statistical analysis been performed appropriately and rigorously?

Reviewer #1: Yes

Reviewer #2: No

3. Have the authors made all data underlying the findings in their manuscript fully available?

Reviewer #1: Yes

Reviewer #2: Yes

4. Is the manuscript presented in an intelligible fashion and written in standard English?

Reviewer #1: Yes

Reviewer #2: No

5. Review Comments to the Author

Reviewer #1: Overall, this is a useful descriptive study of reproductive health needs.

Abstract

States higher risk of STI. And pregnancy?

We did ask questions on pregnancy but this is a topic for another paper. We wanted this paper to mainly focus on sexual health. 

Intro

Can authors site what is known about repro risk in JJ youth? 

In Australia, a survey was conducted among juvenile justice youth detention in New South Wales. Of 19 young girls surveyed, 31.6% had ever been pregnant, first pregnancy was at 14.2 years. No questions were asked on abortions, miscarriages, stillbirths or drugs/smoking/alcohol while pregnant (Source: 2015. Young People in Custody Health Survey: Full Report. Justice Health & Forensic Mental Health Network and Juvenile Justice NSW. NSW Government: Malabar, Sydney.)

Also, any hypothesized differences between kids with low levels of involvement vs longer-term incarceration

Young people with lower levels of criminal justice involvement vs those in detention were associated with lower rates of ever having had sex (Chi square p<0.05). There were no differences between young people with medium levels of criminal justice involvement vs those in detention in terms of ever having had sex (Chi square p>0.05).

Methods

Do not understand why approach necessary. Can authors clarify? Why were the selection criteria chosen? 

We selected young people aged 14-17 years old as there were very few sexual health and behavior studies conducted on young people below 16 years of age in Australia. Sexual health studies become more difficult to progress if the respondent is below 14 years of age due to issues with obtaining parental/guardian consent and also having to ensure that the respondent truly understands what they are volunteering for as a mature minor. 

Whose missing from the sample since its not representative?

We can only compare age, sex and Indigenous status with other data from official government juvenile justice sources in Australia. 

We did not compare the gender of this study population with other juvenile justice sources since we oversampled girls in our study for advanced statistical analysis (Note: 80% are boys under juvenile justice supervision). We compared this study population and age matched them to groups among juvenile defendants and youth justice populations in Queensland and Western Australia. We found that the proportion of young people, aged 14-15 years and 16-17 years in the MeH-JOSH study population, may differ by 2%-6% compared to their community based peers under official government juvenile justice supervision (Figure 1). 

Figure 1. Young people by age group under supervision on an average day by state and compared with MeH-JOSH study sample

Source: AIHW Youth Justice Qld S132a-S132c and WA S134a-S134c. Available at: https://www.aihw.gov.au/getmedia/38637613-61ed-4709-a42a-1c1b77d8138d/aihw-juv-116-state-and-territory-tables-s128-to-s143-2016-17-data-table.xlsx.aspx

Children’s Court of Qld Annual Report 2016-2017

In general, the study had an over representation of Aboriginal (Indigenous) young people when we compared it to the general Aboriginal populations in Australia and reflects the greater involvement of this population in the justice system (Figure 2). 

Figure 2. Aboriginal populations in Australia and MeH-JOSH

Source: *AIHW Youth Justice. Table S31: Australian population aged 10–17 by Indigenous status, states and territories, December 2013 to December 2017. Available at: https://www.aihw.gov.au/getmedia/993d8e77-b18e-4482-b9b9-12fe838db158/aihw-juv-128-youth-detention-population-in-Australia-2018-data-tables.xlsx.aspx

# Australian Bureau of Statistics (2016). 3238.0.55.001 – Estimates of Aboriginal and Torres Strait Islander Australians, June 2016. Available at: http://www.abs.gov.au/ausstats/abs@.nsf/Lookup/3238.0.55.001main+features1June%202016

^2018 Secondary School Student Sexual Health Survey, personal communication, C Fisher.

Exactly 42 mintues? seems awkward. suggest saying 40 min.

42 minutes is more accurate

consent process is stated for only 1 of the recruitment methods

We have added to the manuscript:

“Due to the nature of this population and from our experiences of conducting surveys among young offenders in Australia[3], we expected that a high proportion of young people would not have a good relationship with their parent(s) or with the adults responsible for them. Thus human research ethics approval was sought and given for young people recruited in the survey to be treated as mature minors.[14] Nevertheless, all young people approached were asked by recruiters if we could contact their parents or guardians for permission to allow them to participate in the survey. Almost all respondents refused us contact or to give any contact details of their parents or guardians preferring to give consent themselves. Recruiters were required to administer a Gillick Competency checklist to ensure that all respondents met the criteria of a mature minor.”

Discussion

First sentence very long

Thank you for pointing this out. We have edited the sentence to make our points clearer.

“We determined that young people involved in the justice system are at greater risk for STIs for the following reasons: (i) they have a higher engagement in sex compared with their peers in school surveys (1.3 to 3.2 times higher); (ii) are often starting to have sex at a younger age (median age 14 years compared to 17 years in the general Australian population and 15 years in the general Indigenous population)19 20; (iii) are significantly more likely to report large numbers of sexual partners (6+) at a young age; and (iv) around half were not using a condom during their last sexual encounter.” 

Do authors think youth recalling HPV vaccination correctly? Is never been vaccinated variable category based on records or youth recall?

Memory recall is not altogether an accurate indicator but other studies in Australia also reveal similar results for similar questions on HPV (e.g. SSASH 2018), most do not recall if they had been vaccinated for HPV which is an interesting finding in itself. Perhaps in future studies we can ask respondents for their consent to match them up with their vaccination records.

I'm not Austrialian. Familiar with Aboriginal but what's the significant of Torres Strait Islander? Perhaps explain briefly in Intro of Methods for international audience. I'm guessing these are also "indigenous" youth, but not sure.

Torres Strait Islanders are also Indigenous in Australia. We have added the following:

“Young people who identified as Aboriginal and/or Torres Strait Islander (Indigenous) descent were overrepresented in the survey sample compared with the general population (44% versus 5%), reflecting the greater involvement of this population in the justice system.”

Regarding disproportionate confinement, what is the prevalence of aboriginal and islanders on the island?

According to the Australian Bureau of Statistics, there are 590,062 Aboriginal People and 32,344 Torres Strait Islanders in Australia. In addition, there are 26,767 who are both Aboriginal and Torres Strait Islanders.

https://quickstats.censusdata.abs.gov.au/census_services/getproduct/census/2016/quickstat/IREG307?opendocument

Juvenile justice data

Limitations

Authors' frustration with approval process and resultant sampling framework comes through. perhaps revise

We would like to retain this paragraph as it provides a rationale for the sampling framework.

“A purposive sampling design based on strict selection criteria was used to recruit young people in the community; participants must be aged between 14 and 17 years and had to have had contact with the criminal justice system in the past or present. This approach was necessary as we had been denied permission to conduct a cross-sectional representative survey of young people (both in detention and those serving community orders) from youth justice and corrective services departments in Queensland and Western Australia.” 

I would like to see the authors expand on differences between kids of varying levels of contact. Did they do sub-analyses? At the very least, Discussion should comment on this. Could be added as a limitation

We did not conduct any subanalysis as we wanted this first paper to be descriptive.

Table

Suggest title column with name that clearly indicates that MEH group is the JJ group

Technically, some aren’t under juvenile justice supervision but under supervision of the police etc. We have added, “MEH-JOSH (JUSTICE INVOLVED YOUNG PEOPLE)”

Formatting of tables could be improved. Suggest n and then % in parenthesis (unless authors are following PLOS table formatting guidelines that I am unaware of). It will make table more digestible.

It becomes harder for us to edit n (%) as they are not in separate columns. We leave this to the editor to decide. This format has been accepted in our other publications.

Footnotes should briefly explain what the different samples are.

Added to the tables:

1 2014 YMM – 2014 Young Minds Matter: 2nd Survey of the Mental Health of Australian Children Survey

2 2018 SSASH – 6th National Survey of Secondary Students and Sexual Health

What is the rationale for the bolding? It seems inconsistent.

Fixed

 

Reviewer #2: This study offered a descriptive comparison of sexual behaviors among justice-involved and community youth in Australia. The authors interviewed adolescents currently or previously involved with the criminal justice system, and compared their responses to non-delinquent peers using alternative data sources. Youth in the justice-involved sample were more likely to engage in sex and start having sex at a younger age, and were more likely to report having 6 or more partners.

Overall, I think the premise of this paper is a useful contribution. While researchers argue that justice-involved youth are at heightened risk of sexual risk-taking, there are few formal comparisons with nationally representative samples to empirically support this assertion. Although this paper provides evidence for these assumptions, there are a number of substantial issues that would need to be addressed prior to publication.

Major issues

1. The authors could improve the introduction by making a stronger argument as to why delinquent populations may be at risk for sexual risk-taking. Right now, it seems as though their argument rests on the idea that delinquent populations may not be reached by traditional intervention efforts, and ignores the theoretical and empirical work that suggests that the same factors that put youth at risk for delinquency may drive their tendency to engage in riskier sexual behaviors. I would recommend reviewing a number of articles:

a. Robbins, R. N., & Bryan, A. (2004). Relationships between future orientation, impulsive sensation seeking, and risk behavior among adjudicated adolescents. Journal of adolescent research, 19(4), 428-445.

b. Bryan, A. D., Schmiege, S. J., & Magnan, R. E. (2012). Marijuana use and risky sexual behavior among high-risk adolescents: trajectories, risk factors, and event-level relationships. Developmental psychology, 48(5), 1429.

c. In general, Angela Bryan has a number of relevant articles on this subject that are worth reading.

 

Thank you. We have added the paragraph,

“Studies have suggested that drug addiction and mental health disorders that put youth at risk for offending may also drive their tendency to engage in riskier sexual behaviours.4-7 One study reported increased sexual risk behaviours among juvenile detainees compared to the general population.8 Formal comparisons of young offenders with community populations and their sexual health and behaviours are extremely limited.9”

2. Similarly, although formal comparisons with community populations are limited, there are a few studies that address a similar subject matter that seem like they should be included:

a. Elkington, K. S., Teplin, L. A., Mericle, A. A., Welty, L. J., Romero, E. G., & Abram, K. M. (2008). HIV/sexually transmitted infection risk behaviors in delinquent youth with psychiatric disorders: A longitudinal study. Journal of the American Academy of Child & Adolescent Psychiatry, 47(8), 901-911.

b. Teplin, L. A., Mericle, A. A., McClelland, G. M., & Abram, K. M. (2003). HIV and AIDS risk behaviors in juvenile detainees: Implications for public health policy. American Journal of Public Health, 93(6), 906-912.

c. Teplin, L. A., Elkington, K. S., McClelland, G. M., Abram, K. M., Mericle, A. A., & Washburn, J. J. (2005). Major mental disorders, substance use disorders, comorbidity, and HIV-AIDS risk behaviors in juvenile detainees. Psychiatric Services, 56(7), 823-828.

d. DiClemente, R. J., Lanier, M. M., Horan, P. F., & Lodico, M. (1991). Comparison of AIDS knowledge, attitudes, and behaviors among incarcerated adolescents and a public school sample in San Francisco. American journalof public health,81(5), 628–630.

Thank you. We have added the paragraph,

“Studies have suggested that drug addiction and mental health disorders that put youth at risk for offending may also drive their tendency to engage in riskier sexual behaviours.[4-7] One study reported increased sexual risk behaviours among juvenile detainees compared to the general population.[8] Formal comparisons of young offenders with community populations and their sexual health and behaviours are extremely limited.[9]”

3. The authors should provide more information on what they mean by a “purposive sampling design.” The authors do not explain what the calculated sample size was, which is odd given the heading of the section.

We have edited the paragraph to read more accurately,

“A purposive sampling design based on strict selection criteria was used to recruit young people in the community; participants must be aged between 14 and 17 years and had to have had contact with the criminal justice system in the past or present. This approach was necessary as approval was not granted to conduct a cross-sectional representative survey of young people (both in detention and those serving community orders) from youth justice and corrective services departments in Queensland and Western Australia. 

Quota sample sizes were calculated based on known demographic characteristics of the Australian juvenile offender population (age and gender).[10] Aboriginal young people in this study were over represented (44%) compared to 5% in the general Australian population[13] and reflects the greater involvement of this population in the justice system. In Australia, the majority of young people under supervision on an average day in 2017–18 were male (81%).[10] Females were deliberately oversampled in this study to enable more advanced statistical analysis”

4. The authors should offer some information as to if the sample differed across the four different recruitment methods, given that they are ultimately combined.

We added the following sentence,

Site of recruitment was not a confounding factor for age and gender although it was significant for Aboriginality (p<0.05). 

5. The brevity of the methods makes it difficult to discern how the authors measured sexual behavior. Many of the terms used are quite broad (justice system involvement, sexual history, sexual knowledge) and could encompass a wide variety of topics. The authors should expand on this section and provide more details as to what their measures asked, providing specific examples of the questions.

We did not list the survey questions as there is a word limit for original papers. Nevertheless, the survey questions and measures are listed in the attached tables.

6. The authors describe the comparison sample under “Data Analysis” but it seems like this information should be included in the methods section, in the same section where the justice-system sample is described. We are also given very few details about these comparison samples and some additional information on these separate studies would be helpful, as well as an explanation of why two different studies were employed. If the two studies included different variables, then the authors should describe the different measures covered in each study. 

We have added the following sentence to make it clearer on how the variables were selected. 

“Descriptive statistics were generated using Statistics SPSS 25 from original datasets of the MeH-JOSH survey and age matched with the Young Minds Matter: 2nd Survey of the Mental Health of Australian Children Survey (YMM), a probability sample of young people aged between 4 and 17 years old from 5,500 randomly sampled families in Australia.[15] Separate age matched data analysis was provided by C. Fisher for the 6th National Survey of Secondary Students and Sexual Health (SSASH), a convenience sample of adolescents in the community that had agreed to take part in an online survey in Australia. [16]. 

The self-reported sexual health and behaviours of participants in our survey were compared with their age-matched peers in the YMM[15] and SSASH[16] surveys. Sexual identity proportions were compared using the Chi-squared test. MeH-JOSH sexual health and behavioural questions for this paper were selected based on the equivalence or sameness of questions from the YMM and SSASH survey (for example, MEH-JOSH – “Can you tell me if you have ever had sex?” YMM and SSASH – “Have you ever had sexual intercourse?”).”

Similarly, there is very little information provided on what method of matching was used. It also seems as though the authors only matched on age, and it is unclear if other characteristics (gender, SES, race/ethnicity) were considered in the matching process. Given the possible differences in the two populations (justice involved vs. nationally representative), it is possible that these differences could explain different rates of sexual risk-taking, beyond justice-system involvement.

We did not match for gender as we know from current youth justice statistical data that juvenile offender populations were mostly male (80%) and >50-80% Indigenous. We have added an extra row in Table 1 to include gender proportions in the different studies. Chi2 tests showed that gender and ever having had sex was not significant (p>0.05) in the MeH-JOSH, YMM and SSASH surveys. Drug use, alcohol abuse and mental health disorders, however, were associated with ever having sex in the MeH-JOSH study but this is for another paper.

7. The results appeared a bit disorganized, and as a result were difficult to follow. First, I think it would be helpful for the authors to build on their analytic plan in the Data Analysis section. Although they note the use of Chi Square tests, it would be helpful to explain which variables they are comparing (and in which samples) before jumping into the results. 

We edited the sentence to be clearer,

“Sexual identity proportions were compared using the Chi-squared test.”

The first paragraph under “Sexual health and behaviours” seems like it should be earlier in the methods, not in the results. A similar set of information on the two nationally representative samples would also be helpful.

Descriptive epidemiological papers usually include demographics of the survey population in the results section as a standard. 

a. The authors report differences in sexual orientation by gender, but it is not clear 1) what sample this is referring to (justice involved, nationally representative, both combined) and 2) why a gender comparison was made. It does not seem like this was part of the goal of the paper, and the authors did not make any comparisons between the different samples. It is evident from Table 2 that this information is just referring to the justice-involved sample, but it is not clear why no formal comparisons were made with the SSASH sample.

We have added clarification on which study it was from in the sentence.

“Of the participants, 91% identified as heterosexual, 6% bisexual, and 1% gay or homosexual/lesbian. Young women in the MeH-JOSH study were more likely to identify as bisexual/homosexual/queer than young men, (17% versus 2%, χ²(1)>=28.75, p<.001). Similarly, young women in the SSASH study were also more likely to identify as non-heterosexual (27.6% vs 21.2%, X2(1)=36.35, p<0.001) (cf. C Fisher). Thirty-one per cent of respondents reported they were in a current relationship, either dating or living with their partner. Among 14-15 year old and 16-17 year old age groups, 27% and 34% respectively, were in a relationship at the time of the interview. (Table 2)”

b. The next paragraph, “Of the total sample, 76%...” also needs to be clarified. It would be easier to interpret if the authors provided the percentages for both groups, rather than providing a percentage and then stating it was 1.3-3.2x higher than the other groups. I’m also not sure what the next sentence (“However, this was lower than the 92%...) refers to. Based on Figure 1 it looks like these analyses were split by age, but this is not clear from the methods or results. In addition, it is not clear if the original item combined oral, vaginal and/or anal sex, or if this was combined by the authors. 

We have edited the paragraph to read more clearly,

“Of the total MeH-JOSH sample, 76% (n=348) reported having ever had oral, vaginal and/or anal sex, which was higher than among school aged children in the SSASH (57%) and YMM (24%) surveys (Fig 1 and Table 2). In the MeH-JOSH survey, 74% (n=338) had had vaginal and/or anal sex compared to 45% in the SSASH survey (Figure 1 and Table 2). We noted that these figures were lower than the 92% sexual experience reported elsewhere among young people under the age of 16 years in New South Wales juvenile detention centres (Fig 1).[2] Condom use at the last sexual encounter was 55% for both MeH-JOSH and SSASH respondents compared to 64% among YMM respondents (Table 3)”

It would be preferable to look at these types of sexual behaviors separately, if possible.

In Table 3, we looked at these sexual behaviours (oral, vaginal and anal sex) separately.

c. The next section (“Of the 348 who reported…) also lacks organization. It would be helpful to provide comparable information across samples, and then provide a formal comparison using a statistical test. For example, 76% of youth in the justice-sample reported having sex for the first time at age 14 or younger. What is the comparable statistic for the other sample? 

The YMM survey did not ask about age of sexual initiation and the SSASH survey collected age of sexual initiation as a categorical rather than a continuous variable and thus we cannot compare the data.

The authors instead provide some descriptive information about the justice-sample, but then offer only a select number of comparisons (e.g., how oral sex was less common and vaginal sex was more common among justice-system participants). There is also no formal statistic provided to show whether these differences are statistically significant.

d. The authors indicate that justice-system youth initiated sexual activity at an earlier age, but it is not clear if this was formally tested. What statistical test did they use?

The YMM survey did not ask age of sexual initiation and the SSASH survey collected age of sexual initiation as a categorical rather than a continuous variable and thus we cannot compare or test the data.

We can only discuss our data with a longitudinal study of Australian adults who recalled their first sexual initiation at 17 years of age. 

17. Rissel C, Heywood W, de Visser RO, Simpson JM, Grulich AE, Badcock PB, et al. First vaginal intercourse and oral sex among a representative sample of Australian adults: the Second Australian Study of Health and Relationships. Sex Health. 2014; 11:406-15.

e. For sexual initiation, it is confusing to say 41% of justice-involved participants reported 6+ partners, which was 2.6x higher than the comparison sample. It would be helpful just to provide the percentage for the other samples as well.

We have edited the sentence to read,

“Of those who have had sex, 41% of justice-involved MeH-JOSH participants reported having six or more sexual partners in their lifetime compared to 16% among school aged children in the YMM survey, and higher at all ages (Figure 3).” 

f. The last section (“Of the 348 sexually experienced participants, 37%...”), it appears the authors only provide descriptive information for the justice involved sample. Without a formal comparison it is difficult to discern if these percentages are low, high or average.

We have no formal comparisons but we reported it for any future studies to compare STI rates in similar populations.

g. Overall, the results would need to be entirely restructured to offer descriptive statistics for all samples, and then provide formal comparisons using the appropriate statistical tests.

Around 76% of young respondents in the MeH-JOSH study are already having sex so to compare them ever having sex with other variables (gender and age) may be self-fulfilling. 

8. The discussion does not entirely map onto the results from the paper. For example, the authors argue justice system youth are at greater risk for STIs, but this ignores the finding that their condom use is comparable to the community samples. 

Condom use was for their last sexual encounter rather than lifetime sexual encounter. We cannot discount other high risk sexual practices.

The authors also argue that most had not been tested for HPV, but we have no comparison to the community sample to know if this is something that is particular to justice system populations. It seems like the primary finding is that justice system youth are more likely to have had sex, but given that all sex is not inherently risky, it seems like some of these other statistics are worth discussing (condom use, HPV testing, HIV testing).

We have no formal comparisons but we reported it for any future studies to compare reported HPV vaccination rates in similar populations.

Minor issues

1. On page 4, it should read, “The Mental Health, Sexual Health and Reproductive…study aimed to describe” (rather than describes).

Fixed

2. On the top of page 5, “The main aim of the survey was…” (not were).

Fixed

3. The authors switch back and forth between saying justice involved vs. using the MeH-JOSH acronym. It would be helpful if this was consistent.

Fixed

4. Table 2 is overwhelming to look at, and could be broken down into smaller pieces for easier interpretation.

 We have now separated this into two tables: Table 3 and Table 4.

6. PLOS authors have the option to publish the peer review history of their article (what does this mean?). If published, this will include your full peer review and any attached files.

Do you want your identity to be public for this peer review? For information about this choice, including consent withdrawal, please see our Privacy Policy.

Reviewer #1: No

Reviewer #2: No

---

## [Decision Letter · Decision Letter 1]

2 Oct 2020

PONE-D-19-26106R1

The sexual behaviours of adolescents  aged between 14 and 17 years involved with the juvenile justice system in Australia: a community-based survey

PLOS ONE

Dear Dr. Butler,

Thank you for submitting your manuscript to PLOS ONE. After careful consideration, we feel that it has merit but does not fully meet PLOS ONE’s publication criteria as it currently stands. Therefore, we invite you to submit a revised version of the manuscript that addresses the points raised during the review process.

Please review my comments in the section marked "Additional Editor Comments" below. I have some significant concerns that the changes needed to meet the publication criteria have not been made. I would like to request these revisions again.

We look forward to receiving your revised manuscript.

Kind regards,

Andrea Knittel

Academic Editor

PLOS ONE

Additional Editor Comments (if provided):

Although this manuscript remains a timely and topical piece, many of the concerns expressed by the reviewers have not been addressed. Reviewer #2 has reviewed the revised manuscript, and outlined the persistent issues in the additional review below. Reviewer #1 was not available for repeat review, but on my reading of the responses I have identified several issues. Many of the critiques were addressed with comments in the response, but no changes were actually made in the manuscript. For example, the request to incorporate some discussion of what is known about about reproductive health risk in this population was clearly intended to motivate a change to the manuscript, not just a sentence about the available literature. The clarifications regarding the sampling frame were also intended to be reflected in the manuscript, and the question about participant recall of HPV vaccination addressed in the discussion or some other place. Please review again the critiques from Reviewer #1, and address each in the manuscript. These substantive changes are necessary to meet the publication criteria for PLOS ONE.

Reviewers' comments:

Reviewer's Responses to Questions

**Comments to the Author**

1. If the authors have adequately addressed your comments raised in a previous round of review and you feel that this manuscript is now acceptable for publication, you may indicate that here to bypass the “Comments to the Author” section, enter your conflict of interest statement in the “Confidential to Editor” section, and submit your "Accept" recommendation.

Reviewer #2: (No Response)

2. Is the manuscript technically sound, and do the data support the conclusions?

Reviewer #2: (No Response)

3. Has the statistical analysis been performed appropriately and rigorously? 

Reviewer #2: (No Response)

4. Have the authors made all data underlying the findings in their manuscript fully available?

Reviewer #2: (No Response)

5. Is the manuscript presented in an intelligible fashion and written in standard English?

Reviewer #2: (No Response)

6. Review Comments to the Author

Reviewer #2: The author's response is helpful and the changes largely address my original comments. A few things to consider:

I think including the Chi Square comparisons sets the reader up to think other formal comparisons will be made. I understand this is a descriptive paper, but given no other gender comparisons were made, it seems a bit arbitrary. It would be helpful to understand why these specific comparisons were included, whereas other gender comparisons were not. On page 8 the authors also distinguish between the 14-15 YO and 16-17 YOs in regard to relationship status, but I don't see this separated in Table 2 anywhere.

I still have some concerns that the authors may be overstating findings since formal comparisons were not possible. For example, on page 10 the authors state that "We determined that young people involved in the justice system are at great risk for STIS for the following reasons...around half were not using a condom during their last sexual encounter." As I brought up previously, the rates for the other samples were also not that high, and this phrasing makes it seem as though the justice sample is at a statistically higher risk for STI partly due to condom use (compared to the other samples). I think it would reflect the findings more accurately to explain the findings more descriptively, rather than using language that makes it seem formal comparisons were made.

It seems reasonable to at least include a limitation in the discussion, explicitly stating that formal statistical comparisons could not be made across samples.

7. PLOS authors have the option to publish the peer review history of their article (what does this mean?). If published, this will include your full peer review and any attached files.

Reviewer #2: No

---

## [Author Response · Author response to Decision Letter 1]

13 Nov 2020

Dear Managing Editor and Reviewers,

We thank you for your consideration of our manuscript and for the detailed feedback that has been provided. We have also addressed Reviewer #1’s comments from the first round again as requested.

Substantial changes have been made to the manuscript which we believe has greatly enhanced its rigour and suitability for publication. Please see our responses to specific comments below. We look forward to hearing your response.

Kind regards,

Tony Butler

Editor Comments (if provided):

Although this manuscript remains a timely and topical piece, many of the concerns expressed by the reviewers have not been addressed. Reviewer #2 has reviewed the revised manuscript, and outlined the persistent issues in the additional review below. Reviewer #1 was not available for repeat review, but on my reading of the responses I have identified several issues. Many of the critiques were addressed with comments in the response, but no changes were actually made in the manuscript. For example, the request to incorporate some discussion of what is known about about reproductive health risk in this population was clearly intended to motivate a change to the manuscript, not just a sentence about the available literature. The clarifications regarding the sampling frame were also intended to be reflected in the manuscript, and the question about participant recall of HPV vaccination addressed in the discussion or some other place. Please review again the critiques from Reviewer #1, and address each in the manuscript. These substantive changes are necessary to meet the publication criteria for PLOS ONE.

Reviewer #1: Overall, this is a useful descriptive study of reproductive health needs.

Comment: Abstract

States higher risk of STI. And pregnancy?

Response: We did ask questions on pregnancy and reproductive health but this will be covered in a separate paper. We wanted this paper to focus on the findings of sexual health and sexual behaviours.

Comment: Intro

Can authors site what is known about repro risk in JJ youth?

Response: We have added to the manuscript:

“In Australia, a survey was conducted among juvenile justice youth detention in New South Wales. Of 19 young girls surveyed, 31.6% had ever been pregnant, first pregnancy was at 14.2 years. No questions were asked on abortions, miscarriages, stillbirths or drugs/smoking/alcohol while pregnant.[3]”

Comment: Also, any hypothesized differences between kids with low levels of involvement vs longer-term incarceration

Response: We have added to the manuscript in the Discussion:

“Since this was an exploratory study, we did not hypothesise about differences between the incarcerated youth justice population and those with less involvement. Differences stratified by involvement with the justice system can be explored in future.”

Comment: Methods

Do not understand why approach necessary. Can authors clarify? Why were the selection criteria chosen?

Response: We have added to the manuscript to Sampling and sample size:

“A sample of young people aged 14-17 years old was selected to reflect the age of the youth justice population in Australia. Those over 17 (that is, 18 or older) are considered adults. Ethical issues prevented interviewing those below 14 years, such as obtaining parental/guardian consent, and also having to ensure that very young respondents could truly understand what they would be consenting to.“ 

Comment: Whose missing from the sample since its not representative?

Response: The aim of the study was not to generate a statistically representative sample of all justice-involved young people in QLD and WA (a group for which there is not an obvious sampling frame from which a random sample could be drawn) but to explore the sexual health and behaviours in a diverse range of young people across the spectrum of justice involvement. 

We note that this is one of few studies, and the only recent study, to examine the sexual behaviours amongst this population, which is due to the practical and methodological difficulties of robustly surveying this group of young people.

Regarding the generalisability of the findings, we have added to the manuscript in the Discussion:

“Limitations of this study include the use of a non-random sampling strategy (necessary to overcome practical difficulties of reaching such individuals), resulting in a sample that may not be statistically representative of the target population (i.e. the full range of justice-involved young people).”

Comment: Exactly 42 mintues? seems awkward. suggest saying 40 min.

Response: Agree, this has been changed to approximately 40 minutes. 

Comment: consent process is stated for only 1 of the recruitment methods

Response: We have added the following text to explain the consent process as part of a separate section Consent:

“Due to the nature of this population and from our experiences of conducting surveys among young offenders in Australia[3], the study expected that a high proportion of young people would not have a good relationship with their parent(s) or with the adults responsible for them. Thus human research ethics approval was sought and given for young people recruited in the survey to be treated as mature minors.[14] Nevertheless, all young people approached outside the courts were still asked by recruiters if we could contact their parents or guardians for permission to allow them to participate in the survey. Almost all respondents refused us contact or to give any contact details of their parents or guardians preferring to give consent themselves. Recruiters were required to administer a Gillick Competency checklist to ensure that all respondents met the criteria of a mature minor.” 

Comment: Discussion

First sentence very long

Response: Thank you for pointing this out. We have edited the sentence to make our points clearer.

“Young people involved in the justice system have a higher engagement in sex compared with their peers in school surveys (1.3 to 3.2 times higher), are often starting to have sex at a younger age (median age 14 years compared to 17 years in the general Australian population and 15 years in the general Indigenous population)[17] [18], and are significantly more likely to report large numbers of sexual partners (6+) at a young age. While similar rates of young people used a condom during their last sexual encounter across MeH-JOSH, SSASH and YMM, engagement in more frequent and riskier sexual behaviours suggests that this may pose more of a STI risk to justice-involved young people. 

Comment: Do authors think youth recalling HPV vaccination correctly? Is never been vaccinated variable category based on records or youth recall?

Response: We have added to the manuscript in the Discussion:

“Findings are based on self-report and may be impacted by recall or social desirability bias. While memory recall is not altogether an accurate indicator other studies in Australia also reveal similar results for similar questions on HPV (e.g. SSASH 2018), most do not recall if they had been vaccinated for HPV which is an interesting finding in itself. Perhaps in future studies we can ask respondents for their consent to match them up with their vaccination records.”

Comment: I'm not Austrialian. Familiar with Aboriginal but what's the significant of Torres Strait Islander? Perhaps explain briefly in Intro of Methods for international audience. I'm guessing these are also "indigenous" youth, but not sure.

Response: Torres Strait Islanders are also Indigenous in Australia. We have added “(Indigenous)” to clarify in the manuscript:

“Young people who identified as Aboriginal and/or Torres Strait Islander (Indigenous) descent were overrepresented in the survey sample compared with the general population (44% versus 5%), reflecting the greater involvement of this population in the justice system.”

Comment: Regarding disproportionate confinement, what is the prevalence of aboriginal and islanders on the island?

Response: We have clarified in our manuscript:

“Aboriginal and/or Torres Strait Islander (Indigenous) young people in this study sample were over represented (44%) compared to 5% in the general Australian population and this reflects the greater involvement of this population in the justice system.[13]”

Comment: Juvenile justice data

Limitations

Authors' frustration with approval process and resultant sampling framework comes through. perhaps revise

Response: We have revised this in the manuscript:

“A purposive sampling design based on strict selection criteria was used to recruit young people in the community; participants must be aged between 14 and 17 years and had to have had contact with the criminal justice system in the past or present. This approach was necessary as permission was not granted to conduct the survey with young people in detention and those serving community orders.” 

Comment: I would like to see the authors expand on differences between kids of varying levels of contact. Did they do sub-analyses? At the very least, Discussion should comment on this. Could be added as a limitation

Response: We have added to the manuscript in the Discussion:

“Since this was an exploratory study, we did not hypothesise about differences between the incarcerated youth justice population and those with less involvement. Differences stratified by involvement with the justice system can be explored in future.”

Comment: Table

Suggest title column with name that clearly indicates that MEH group is the JJ group

Response: To clarify, we have edited the table titles as follows:

Comment: Formatting of tables could be improved. Suggest n and then % in parenthesis (unless authors are following PLOS table formatting guidelines that I am unaware of). It will make table more digestible.

Response: Fixed.

Comment: Footnotes should briefly explain what the different samples are.

Response: Added to the tables:

1 2014 YMM – 2014 Young Minds Matter: 2nd Survey of the Mental Health of Australian Children Survey

2 2018 SSASH – 6th National Survey of Secondary Students and Sexual Health

Comment: What is the rationale for the bolding? It seems inconsistent.

Response: Fixed.

Reviewers' comments:

Reviewer's Responses to Questions

Comments to the Author

1. If the authors have adequately addressed your comments raised in a previous round of review and you feel that this manuscript is now acceptable for publication, you may indicate that here to bypass the “Comments to the Author” section, enter your conflict of interest statement in the “Confidential to Editor” section, and submit your "Accept" recommendation.

Reviewer #2: (No Response)

2. Is the manuscript technically sound, and do the data support the conclusions?

Reviewer #2: (No Response)

3. Has the statistical analysis been performed appropriately and rigorously? 

Reviewer #2: (No Response)

4. Have the authors made all data underlying the findings in their manuscript fully available?

Reviewer #2: (No Response)

5. Is the manuscript presented in an intelligible fashion and written in standard English?

Reviewer #2: (No Response)

6. Review Comments to the Author

Reviewer #2: The author's response is helpful and the changes largely address my original comments. A few things to consider:

Comment: I think including the Chi Square comparisons sets the reader up to think other formal comparisons will be made. I understand this is a descriptive paper, but given no other gender comparisons were made, it seems a bit arbitrary. It would be helpful to understand why these specific comparisons were included, whereas other gender comparisons were not. 

Response: Agreed. Gender comparisons were analysed for all variables and all tables have included these p-values. We have added to the manuscript:

“The self-reported sexual health and behaviours of participants in the MEH-JOSH survey (including sexual identity and attraction, relationship status, ever having sex (any, vaginal, anal, oral), sexual initiation, lifetime sexual encounters, last sexual encounter, STI testing and diagnosis, HPV vaccination, and willingness to have HPV vaccine) were compared with their age-matched peers in the YMM[15] and SSASH[16] surveys where applicable. Chi-squared tests were used to analyse gender differences in these variables in the MEH-JOSH survey. A test of two proportions was used to analyse gender differences in types of offences committed.”

The following gender comparison findings were added to the Participants section (please see Participants section in the manuscript to see full integration of these findings):

• There was a higher proportion of males than females in the 16-17 age group (56% versus 44%, χ²(1)=5.683, p=.017).

• A higher proportion of males were on a current sentence or order compared to females (40% versus 30%, χ²(1)=4.567, p=.033).

The following gender comparison findings were added to the Results section (please see Results section in the manuscript to see full integration of these findings): 

• Young females in the MeH-JOSH study were more likely to identify as bisexual than young males, (15% versus 1%, χ²(3)=34.295, p<.001), and this is also reflected in their sexual attraction (14% versus 3%, χ²(3)=20.031, p<.001). This is similar to the SSASH study whereby young females in the general population were also more likely to identify as bisexual (C Fisher, personal communication, September 2020).

• Justice-involved young females were more likely to have had sex with a person 18 years or older for their first sexual partner (13% versus 3%, χ²(1)=12.056, p=.001), and their last sexual partner (32% versus 12%, χ²(1)=19.812, p<.001), compared to males.

• Justice-involved young males were more likely to report numbers of 6 or more sexual partners than females (45% versus 34%, χ²(1)=4.091, p=.043).

• For their last sexual encounter, justice-involved young females were more likely to have sex with their current girlfriend/boyfriend (54% versus 40%), while males were more likely to have sex with a stranger (16% versus 4%, χ²(3)=13.629, p=.003). Young males were more likely to use a condom during their last sexual encounter than females (59% versus 48%, χ²(1)=4.094, p=.043).

• Of the 348 sexually experienced participants, 37% (n=129) reported prior testing for sexually transmissible infections (STIs) with young females more likely to have been tested recently in the last year compared to males (44% versus 24%, χ²(2)=9.392, p=.009). Young females were more likely to be tested by a general practitioner (57% versus 40%), while males were more likely to be tested in a juvenile detention centre or prison (20% versus 3%, χ²(5)=19.377, p=.002).

• Young females were more likely to have been HPV vaccinated than males (38% versus 26%, χ²(2)=7.361, p=.025).

• A higher proportion of young females than males indicated a willingness to be HPV vaccinated (68% versus 53%, χ²(2)=6.988, p=.030).

Comment: On page 8 the authors also distinguish between the 14-15 YO and 16-17 YOs in regard to relationship status, but I don't see this separated in Table 2 anywhere.

Response: Well-spotted. We have modified the sentence to remove the reference to Table 2 here in our manuscript as follows:

 “Among 14-15 year old and 16-17 year old age groups, 27% and 34% respectively, were in a relationship at the time of the interview.”

Comment: I still have some concerns that the authors may be overstating findings since formal comparisons were not possible. For example, on page 10 the authors state that "We determined that young people involved in the justice system are at great risk for STIS for the following reasons...around half were not using a condom during their last sexual encounter." As I brought up previously, the rates for the other samples were also not that high, and this phrasing makes it seem as though the justice sample is at a statistically higher risk for STI partly due to condom use (compared to the other samples). I think it would reflect the findings more accurately to explain the findings more descriptively, rather than using language that makes it seem formal comparisons were made.

Response: We agree with this concern of overstating findings, and have modified the language in our manuscript as follows:

“We have observed that young people involved in the justice system have a higher engagement in sex compared with their peers in school surveys (1.3 to 3.2 times higher), are often starting to have sex at a younger age (median age 14 years compared to 17 years in the general Australian population and 15 years in the general Indigenous population)[17] [18], and are significantly more likely to report large numbers of sexual partners (6+) at a young age. While similar rates of young people used a condom during their last sexual encounter across MeH-JOSH, SSASH and YMM, engagement in more frequent and riskier sexual behaviours suggests that this may pose more of a STI risk to justice-involved young people.” 

“The study demonstrates how young people involved in the justice system may be at high risk of STIs and that it is possible to target this usually hard to access population through youth oriented community spaces without juvenile justice or corrective services involvement.”

Also, we have modified the language in our abstract:

“The sexual behaviours of young people involved in the justice system in this study suggest they may be at a greater risk for sexually transmissible infections than their age-matched peers in the general population.”

Comment: It seems reasonable to at least include a limitation in the discussion, explicitly stating that formal statistical comparisons could not be made across samples.

Response: We agree. We have added to the manuscript in the Discussion:

“Another limitation of this study is that it was not possible to conduct formal comparisons across these different survey samples.”

 7. PLOS authors have the option to publish the peer review history of their article (what does this mean?). If published, this will include your full peer review and any attached files.

Do you want your identity to be public for this peer review? For information about this choice, including consent withdrawal, please see our Privacy Policy.

Reviewer #2: No

---

## [Editor Report · Decision Letter 2]

25 Nov 2020

The sexual behaviours of adolescents  aged between 14 and 17 years involved with the juvenile justice system in Australia: a community-based survey

PONE-D-19-26106R2

Dear Dr. Butler,

We’re pleased to inform you that your manuscript has been judged scientifically suitable for publication and will be formally accepted for publication once it meets all outstanding technical requirements. Thank you for your thoughtful revisions to the manuscript.

Kind regards,

Andrea Knittel

Academic Editor

PLOS ONE
---

## [Editor Report · Acceptance letter]

10 Dec 2020

PONE-D-19-26106R2 

The sexual behaviours of adolescents aged between 14 and 17 years involved with the juvenile justice system in Australia: a community-based survey 

Dear Dr. Butler:

I'm pleased to inform you that your manuscript has been deemed suitable for publication in PLOS ONE. Congratulations! Your manuscript is now with our production department. 

Kind regards, 

on behalf of

Dr. Andrea Knittel 

Academic Editor

PLOS ONE